# Explore-on-Graph: Incentivizing Autonomous Exploration of Large Language Models on Knowledge Graphs with Path-refined Reward Modeling

**Shiqi Yan**[1,2], **Yubo Chen**[1], **Ruiqi Zhou**[2], **Zhengxi Yao**[2], **Shuai Chen**[4*], **Tianyi Zhang**[4],
**Shijie Zhang**[4], **Wei-Qiang Zhang**[1,2], **Yongfeng Huang**[1,2], **Haixin Duan**[1,3], **Yunqi Zhang**[1*]

[1]Zhongguancun Laboratory
[2]Department of Electronic Engineering, Tsinghua University
[3]Institute for Network Sciences and Cyberspace, Tsinghua University
[4]Ant International, Ant Group

## Abstract

The reasoning process of Large Language Models (LLMs) is often plagued by hallucinations and missing facts in question-answering tasks. A promising solution is to ground LLMs' answers in verifiable knowledge sources, such as Knowledge Graphs (KGs). Prevailing KG-enhanced methods typically constrained LLM reasoning either by enforcing rules during generation or by imitating paths from a fixed set of demonstrations. However, they naturally confined the reasoning patterns of LLMs within the scope of prior experience or fine-tuning data, limiting their generalizability to out-of-distribution graph reasoning problems. To address this issue, in this paper, we propose Explore-on-Graph (EoG), a novel framework that encourages LLMs to autonomously explore a more diverse reasoning space on KGs. To incentivize exploration and discovery of novel reasoning paths, we propose to introduce reinforcement learning during training, whose reward is the correctness of the reasoning paths' final answers. To enhance the efficiency and meaningfulness of the exploration, we propose to incorporate path information as additional reward signals to refine the exploration process and reduce futile efforts. Extensive experiments on five KGQA benchmark datasets demonstrate that, to the best of our knowledge, our method achieves state-of-the-art performance, outperforming not only open-source but also even closed-source LLMs[1].

## 1 Introduction

Large language models (LLMs) have demonstrated impressive capabilities in various natural language processing tasks (Brown et al., 2020; Team et al., 2023; Liu et al., 2024; Dubey et al., 2024). Despite their success, LLMs often struggle to produce faithful reasoning in question-answering (QA) tasks (Wang et al., 2023; Radhakrishnan et al., 2023), primarily due to knowledge gaps and their susceptibility to hallucination (Ji et al., 2023; Guan et al., 2024). To this end, the reliability of LLMs in real-world QA systems remains a significant concern.

To mitigate these issues, a promising approach is to ground LLMs' responses in external and verifiable knowledge sources (Lewis et al., 2020; Li et al., 2023; Ren et al., 2025). Knowledge graphs (KGs), which emerge as an ideal candidate, provide solid factual information to validate and guide the LLMs' reasoning process (Luo et al., 2024; Zhu et al., 2024).

Recent KG-enhanced reasoning methods can be broadly classified into two paradigms: rule-based methods and imitation-based methods. The former (Sun et al., 2023; Zhang et al., 2025) typically leveraged a pre-defined set of rules to ensure the logical consistency and faithfulness during the LLM's reasoning process. The latter (Zhang et al., 2022a; Wu et al., 2024; Jiang et al., 2025) mainly

---

[*]Corresponding author. Contact: `zhangyq@mail.zgclab.edu.cn`

[1]Code and data are available at: https://github.com/ysq111333/EoG

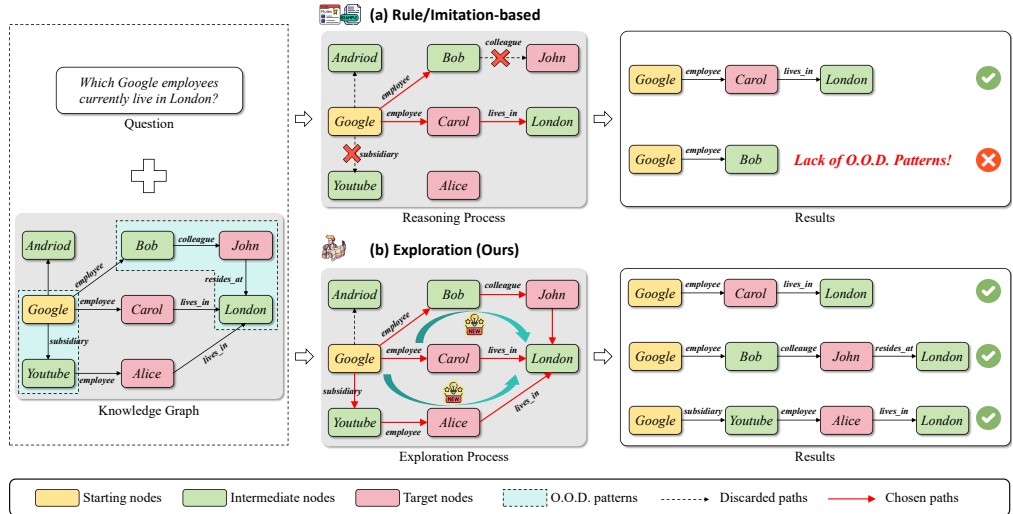

Figure 1: Examples of (a) rule/imitation-based method and (b) our exploration method. O.O.D. stands for Out-Of-Distribution.

trained LLMs on extensive datasets to directly mimic the reasoning patterns and heuristics embedded in the demonstrations.

However, existing methods usually failed to generalize to complex graphs, especially to novel reasoning patterns that fall outside the distribution of the fine-tuning data or pre-defined rules. For example, as shown in Figure 1 (a), "$Google \xrightarrow{employee} Carol \xrightarrow{lives\_in} London$" is the most common reasoning pattern corresponding to the question, and thus can be successfully recognized by existing methods. However, the patterns "$Google \xrightarrow{employee} Bob \xrightarrow{colleague} John \xrightarrow{resides\_at} London$" and "$Google \xrightarrow{subsidiary} Youtube \xrightarrow{employee} Alice \xrightarrow{lives\_in} London$" involve additional steps "$\xrightarrow{colleague}$" and "$\xrightarrow{subsidiary}$" that deviate from the most common pattern, consequently rendering them out-of-distribution and challenging for existing methods. This observation motivates us to believe that navigating beyond the training distribution requires the capability to venture into unfamiliar regions of the graph. For example, in Figure 1(b), although the "$\xrightarrow{colleague}$" and "$\xrightarrow{subsidiary}$" patterns are out of distribution, actively exploring these unfamiliar regions effectively helps discover these novel paths and give more accurate answers. We therefore argue that a complementary capability of autonomous exploration is essential to enhance generalization in the graph reasoning task.

In this paper, we propose a novel framework named Explore-on-Graph (EoG) to incentivize autonomous exploration of LLMs on KGs. Inspired by recent advances in large reasoning models (Guo et al., 2025), we propose to introduce reinforcement learning during training, which is preceded by a Supervised Fine-Tuning (SFT) step, to stimulate exploration capabilities. We use the correctness of the reasoning paths' final answers as reward signals, which can be computed programmatically given the golden label. Moreover, to improve the efficiency and semantic meaningfulness of the exploration process, we propose to incorporate path information as additional reward signals to refine the exploration strategy. Specifically, we introduce a second dedicated training stage and leverage our proposed path-refined reward. To evaluate the effectiveness of our approach, we conducted extensive experiments on five KGQA benchmark datasets. Experimental results demonstrate that our method achieves state-of-the-art performance, substantially outperforming a range of baseline methods and even surpassing the results of powerful closed-source models.

## 2 RELATED WORK

Knowledge graphs, which consist of relational triples that describe real-world entities and their relationships (Hogan et al., 2021; Chen et al., 2022; Zhang et al., 2022b), have been extensively adopted to guide the reasoning process of large language models to improve factuality and reduce hallu-

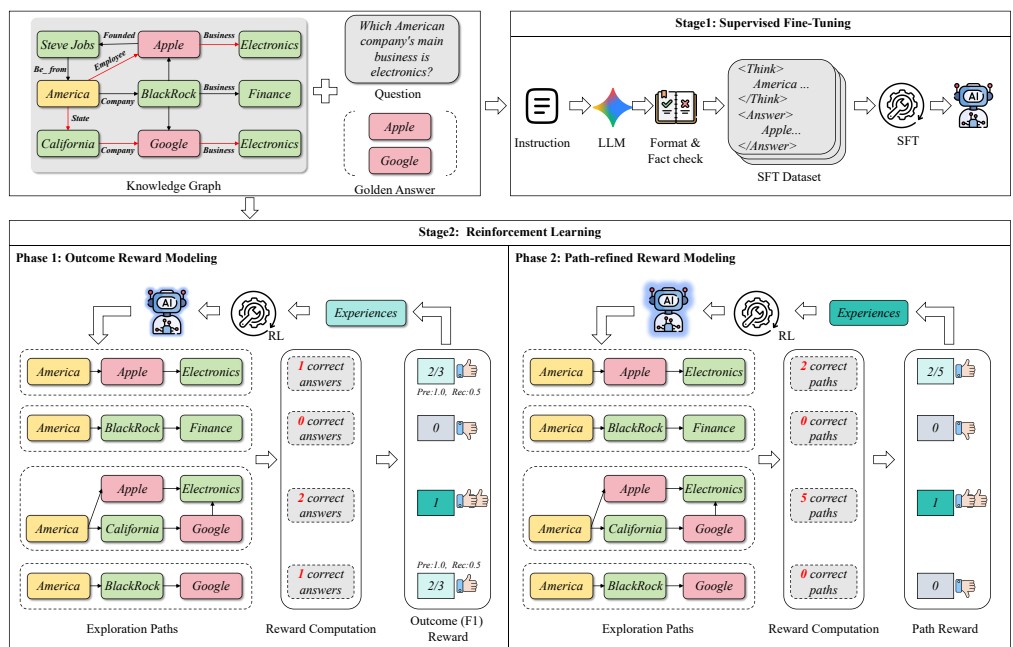

Figure 2: The overall framework of our approach. Red arrows in the knowledge graph stands for the golden reasoning paths of the given question.

cinations (Agrawal et al., 2024). Recent approaches to KG-enhanced reasoning could be broadly categorized into two paradigms based on the generation of the reasoning process: rule-based methods (Sun et al., 2023; Zhang et al., 2025; Li et al., 2025) and imitation-based methods (Wu et al., 2024; Mavromatis & Karypis, 2025; Jiang et al., 2025). To ensure the faithfulness of the reasoning process, rule-based methods guide the LLM's reasoning process on the KGs by pre-defined rules. For example, ToG (Sun et al., 2023) introduced intuitive instructions to prompt LLMs to prune entities and relations from subgraphs. To improve retrieval efficiency, ReKnoS (Wang et al., 2025) proposed to recognize super-relations that are defined as a set of semantically related relations. Moreover, DoG (Li et al., 2025) adopted graph-aware constrained decoding with structured chains to constrain the decoding process. The approaches were generally training-free, but did not enhance the intrinsic reasoning capabilities of the LLMs themselves.

Imitation-based methods focus on emulating reasoning patterns derived from fine-tuning data. Early work mainly tried to convert questions into executable logical forms (e.g. SPARQL) for knowledge graph retrieval (Lan & Jiang, 2020; Das et al., 2021; Ye et al., 2022), which heavily relied on the quality of generated queries. More recent work widely used Chain of Thought (CoT) to enhance LLM reasoning (Wu et al., 2024; Zhao et al., 2024). To reduce incorrect thoughts, RoG (Luo et al., 2024) proposed a planning-retrieval-reasoning framework that grounds plans in KGs. PoG (Chen et al., 2024) further improved the planning process through a reflection mechanism for self-correction. Though Kg-Agent (Jiang et al., 2025) employed several agents to iteratively reason over KGs, it relied on supervised fine-tuning with synthesized program data and failed to generalize beyond pre-defined tool-based reasoning paths.

Different from previous work, we propose to incentivize autonomous exploration via reinforcement learning with path-refined reward modeling, enabling the model to explore novel reasoning paths that fall outside the distribution of pre-defined rules or supervised fine-tuning data. Experimental results on several benchmark datasets prove the effectiveness of our method.

## 3 METHODOLOGY

In this section, we introduce our Explore-on-Graph (EOG) in detail, which consists of a Supervised Fine-Tuning (SFT) stage and a Reinforcement Learning (RL) stage. To equip LLMs with the foundational ability for graph exploration, we introduce the SFT on knowledge graph reasoning tasks in

Section 3.1. To motivate the exploration of various reasoning paths and enhance its efficiency and meaningfulness, we introduce the RL with path-refined reward modeling in Section 3.2. The general framework of our approach is illustrated in Figure 2.

## 3.1 SUPERVISED FINE-TUNING

Directly instructing the LLMs to explore KGs presents significant challenges, including an intractably large action space and extreme reward sparsity (Xiong et al., 2017; Lin et al., 2018), due to lack of prior knowledge. To bridge this gap, CoT offers a pathway of structured reasoning demonstrations, by teaching LLMs the complex multi-step logic necessary for knowledge graph exploration (Wu et al., 2024). Compared to short CoT, long CoT allows more comprehensive and in-depth reasoning by exploring a wider range of logical paths (Chen et al., 2025). Therefore, we employ the long CoT Supervised Fine-Tuning paradigm to establish a foundation for the subsequent exploration phase, which consists of two parts: (1) constructing the long CoT reasoning dataset and (2) fine-tuning LLMs with the constructed dataset.

**Long CoT Dataset Construction.** We carefully design the prompt to generate long CoT for graph reasoning, which requires the reasoning process to be structured, logical and to be aligned with KGs. The detailed prompt could be found in Figure 9. Due to its advanced deep thinking capabilities, we select Gemini 2.5 Flash (Comanici et al., 2025) to perform knowledge distillation to generate long-chain reasoning thoughts and the final answers. Given a knowledge graph $\mathcal{G}$ and a question $q$ as the query input, the generation sample from Gemini 2.5 Flash can be defined as the reasoning process $z$, which comprises both the reasoning paths (within "$< think >< /think >$" tags) and the final answers (within "$< answer >< /answer >$" tags). Then we incorporate additional rules to filter the reasoning process that is both structurally and factually correct. Finally, the SFT dataset $\mathcal{D}_{\text{CoT}}$ is constructed, with high-quality long CoT reasoning processes.

**Supervised Fine-Tuning.** Following dataset construction, we perform SFT to endow the model with structured long CoT reasoning capabilities. We introduce a standard language modeling objective:

$$\mathcal{L}_{\text{SFT}} = -\frac{1}{|\mathcal{D}_{\text{CoT}}|} \sum_{i=1}^{|\mathcal{D}_{\text{CoT}}|} \sum_{j=1}^{|z_i|} \log P(z_{i,j}|z_{i,<j}, \mathcal{G}_i, q_i; \theta), \quad (1)$$

where $(\mathcal{G}_i, q_i, z_i) \in \mathcal{D}_{\text{CoT}}$ is the $i$-th sample in the long CoT dataset, $z_i$ represents the complete reasoning process for the $i$-th sample, $z_{i,j}$ is the $j$-th token of that sequence, and $\theta$ denotes the LLM parameters. Through this phase, the LLM acquires a strong capacity for structured reasoning, establishing an essential foundation for the subsequent exploration phase. Reinforcement Learning (RL) can then effectively refine and diversify these learned pathways based on environmental feedback, rather than starting from unguided random exploration.

## 3.2 REINFORCEMENT LEARNING

While the SFT equips the LLM with the ability to generate structured reasoning paths, its outputs are usually confined to the patterns observed in the training data, potentially leading to suboptimal reasoning. To facilitate the discovery of a broader and more effective exploration space, we structure our reinforcement learning approach into two distinct phases. Inspired by advances such as Deepseek-R1 (Shao et al., 2024), firstly, we introduce the Group Relative Policy Optimization (GRPO) to optimize the exploration policy on KGs based on the correctness of the reasoning path's final answers in Section 3.2.1. Then we integrate a path-refined reward that provides additional signals to steer exploration toward more efficient and meaningful paths in Section 3.2.2.

### 3.2.1 REINFORCEMENT LEARNING WITH OUTCOME REWARD

In the first phase, we optimize the exploration policy $\pi_\theta$ for enhancing the discovery of correct answers via a GRPO-based and outcome-supervised reinforcement learning objective $\mathcal{J}_{\text{GRPO}}(\theta)$. Given the environment $(\mathcal{G}, q) \in \mathcal{D}_{\text{GRPO}}$, our LLM after the SFT stage could generate a group of $S$ candidate exploration processes $\{p_i\}_{i=1}^{S} \subseteq \mathcal{P}_{(\mathcal{G},q)}$, where each $p_i$ is $i$-th process sampled from the old policy $\pi_{\theta_{\text{old}}}$. Similar to a reasoning process $z$, each $p$ contains both the exploration paths and the final answers with structured tags. The policy model could be optimized by maximizing the following reinforcement learning objective $\mathcal{J}_{\text{GRPO}}(\theta)$:

$$\mathcal{J}_{\text{GRPO}}(\theta) = \mathbb{E}_{[\mathcal{G}, q \sim \{\mathbb{P}(\mathcal{G}, q) | (\mathcal{G}, q) \in \mathcal{D}_{\text{GRPO}}\}, \{p_i\}_{i=1}^{S} \sim \pi_{\theta_{\text{old}}}(\mathcal{P}_{(\mathcal{G}, q)} | p; \mathcal{G})]}$$

$$\frac{1}{S} \sum_{i=1}^{S} \frac{1}{|p_i|} \sum_{t=1}^{|p_i|} \left\{ \min \left[ \psi_\theta(p_{i,t}) \hat{A}(p_i), \text{clip}\left(\psi_\theta(p_{i,t}), 1 \pm \epsilon\right) \hat{A}(p_i) \right] - \beta \mathbb{D}_{\text{KL}}(\pi_\theta || \pi_{\text{ref}}) \right\}, \tag{2}$$

where $\psi_\theta(p_{i,t}) = \frac{\pi_\theta(p_{i,t}|q, \mathcal{G}, p_{i<t})}{\pi_{\theta_{\text{old}}}(p_{i,t}|q, \mathcal{G}, p_{i<t})}$, and $\hat{A}(p_i) = \frac{R(p_i) - \text{mean}\left(\{R(p_j)\}_{j=1}^{S}\right)}{\text{std}\left(\{R(p_j)\}_{j=1}^{S}\right)}$. The importance sampling ratio $\psi_\theta(p_{i,t})$ measures how much more likely the new policy $\pi_\theta$ is to generate $p_{i,t}$ compared to the old policy $\pi_{\theta_{\text{old}}}$. $\hat{A}(p_i)$ is the relative normalized advantage. $\epsilon$ and $\beta$ are hyperparameters. $\mathbb{D}_{\text{KL}}(\pi_\theta || \pi_{\text{ref}})$ computes the KL-divergence between the updated policy and a reference policy. $R(p_i)$ is the reward function that measures the quality of diverse exploration paths $\{p_i\}_{i=1}^{S}$.

By optimizing Equation (2), the LLM learns to favor exploration paths that are highly rewarding and reliable relative to other sampled paths. Therefore, we define the outcome reward $R(p)$ to encourage generating exploration paths that contain the right answers. Specifically, we utilize the entity-level F1 score to measure the correctness of the final answers of a generated exploration sequence $p$. We first recognize the set of predicted answer entities $A_p$ through extracting the texts within the "$< answer >< /answer >$" tags in the generated exploration sequence. Considering the set of ground-truth answer entities $A_g$, then the outcome reward $R_{\text{outcome}}(p)$ is calculated as follows:

$$R_{\text{outcome}}(p) = 2 \cdot \frac{Pre \cdot Rec}{Pre + Rec}, \text{where } Pre = \frac{|A_p \cap A_g|}{|A_p|}, \quad Rec = \frac{|A_p \cap A_g|}{|A_g|} \tag{3}$$

We set the reward to 0 if the predicted answer set $A_p$ is empty. Note that the outcome reward $R_{\text{outcome}}(p)$ is automatically set to 0 for all samples that do not contain correctly formatted answer tags in the exploration sequence. Therefore, this reward implicitly encourages the correct structured exploration processes during the policy optimization process. For this reason, we did not include an additional format reward in this phase.

### 3.2.2 REINFORCEMENT LEARNING WITH PATH-REFINED REWARD

The process of generating an exploration path reflects the LLM's reasoning trajectory through a KG. Rewarding correct paths minimizes exploration of wrong directions. In addition, the exploration paths with sequential chains of connected relational triples contain rich semantic information, which can help the LLM generate more comprehensive and in-depth exploration processes. In this phase, we suggest using path information to generate auxiliary rewards, thereby optimizing the exploration process and curtailing inefficient actions.

The path-based reward, $R_{\text{path}}(p)$, is formulated to quantify the proportion of ground-truth reasoning steps successfully articulated in the generated thought process. To obtain the ground-truth reasoning path $r_g$ for datasets that do not provide them explicitly, we implement a "Search-and-Verify" pipeline. First, We identify the topic entities in the question and the ground-truth answer entities in the Knowledge Graph. We then perform a Breadth-First Search (BFS) to retrieve all potential paths connecting the topic entities to the answer entities within a maximum hop constraint. This step ensures high recall of potential reasoning chains. Second, since BFS may yield spurious paths that are topologically connected but semantically unrelated, we employ a LLM (e.g., Gemini-2.5-Flash) to semantically verify these paths. The LLM is prompted to determine if a path logically corresponds to the question's intent, and only validated paths are retained as $r_g$.

Given a ground-truth reasoning path $r_g$, which consists of a set of structured triplets, we first extract this set of triplets, denoted as $T = \{(s_i, r_i, o_i)\}_{i=1}^{|T|}$. For each triplet $t_i = (s_i, r_i, o_i) \in T$, we check if all three components—subject $s_i$, relation $r_i$, and object $o_i$—are present as substrings within the generated reasoning text $p_{\text{think}}$. The reward is then calculated as the ratio of fully matched triplets to the total number of triplets in the ground-truth path:

$$R_{\text{path}}(p) = \frac{1}{|T|} \sum_{t_i \in T} \mathbb{I}(s_i \in p_{\text{think}} \wedge r_i \in p_{\text{think}} \wedge o_i \in p_{\text{think}}) \tag{4}$$

where $\mathbb{I}(\cdot)$ is the indicator function, which returns 1 if the condition is true and 0 otherwise. This function provides a direct and interpretable score of how faithfully the model's reasoning follows the correct path, hence contributing to the efficiency and meaningfulness of exploration process.

Table 1: Performance of EoG and previous state-of-the-art models on the five KGQA test sets. The best scores are in bold. $^\dagger$ marks the performance of the closed-source model which uses the same input as our EoG. $\text{EoG}_{SFT}$ is the model which is only trained with SFT datasets but not with RL.

| Method | Model | WebQSP | | CWQ | | GrailQA | | QALD10-en | | 2WikiMultihop | |
|---|---|---|---|---|---|---|---|---|---|---|---|
| | | Hit@1 | F1 | Hit@1 | F1 | Hit@1 | F1 | Hit@1 | F1 | Hit@1 | F1 |
| KD-CoT (2023) | Llama-2-7B | 68.6 | 52.5 | 55.7 | - | - | - | - | - | - | - |
| EWEK-QA (2024) | - | 71.3 | - | 52.5 | - | 60.4 | - | - | - | - | - |
| ToG (2023) | ChatGPT | 76.2 | - | 57.6 | - | 68.7 | - | 50.2 | - | - | - |
| | GPT-4 | 82.6 | - | 68.5 | - | 81.4 | - | 53.8 | - | - | - |
| RoG (2024) | Llama-2-7B | 85.7 | 70.8 | 62.6 | 56.2 | - | - | - | - | - | - |
| ODA (2024) | GPT-4 | - | - | - | - | - | - | 66.7 | - | - | - |
| EffiQA (2025) | GPT-4 | 82.9 | - | 69.5 | - | 78.4 | - | 51.4 | - | - | - |
| GNN-RAG (2025) | Llama-2-7B | 85.7 | 71.3 | 66.8 | 59.4 | - | - | - | - | - | - |
| KG-Agent (2025) | Llama-2-7B | 83.3 | 81.0 | 72.2 | 69.8 | - | 86.1 | - | - | - | - |
| DoG (2025) | Qwen2.5-7B | 92.7 | - | 74.1 | - | - | - | - | - | 84.2 | - |
| | Llama-3.1-8B | 91.4 | - | 76.2 | - | - | - | - | - | 84.1 | - |
| GCR (2025) | Llama-3.1-8B | 92.2 | 79.1 | 75.8 | 61.7 | - | - | - | - | - | - |
| $^\dagger$Gemini-2.5 Flash | - | 91.8 | 78.2 | 65.5 | 59.3 | 90.3 | 83.8 | 56.7 | 46.2 | 83.9 | 83.1 |
| $^\dagger$Gemini-2.5 Pro | - | 92.1 | 79.8 | 71.9 | 65.3 | 91.6 | 84.5 | 58.6 | 48.3 | 85.1 | 82.6 |
| $^\dagger$GPT-5 | - | 86.1 | 77.5 | 74.1 | 67.6 | 90.5 | 85.4 | 59.2 | 50.4 | 84.2 | 83.4 |
| $\text{EoG}_{SFT}$ | Qwen2.5-7B | 83.9 | 72.6 | 68.3 | 60.3 | 89.2 | 87.6 | 55.6 | 44.1 | 82.5 | 81.9 |
| | Llama-3.1-8B | 86.3 | 74.5 | 70.5 | 62.1 | 91.4 | 88.2 | 57.1 | 48.7 | 83.1 | 82.7 |
| EoG | Qwen2.5-7B | 90.7 | 78.1 | 82.7 | 73.8 | 91.7 | 88.5 | 67.3 | 57.8 | 83.9 | 82.9 |
| | Llama-3.1-8B | **92.8** | **81.3** | **86.6** | **77.9** | **92.1** | **90.6** | **70.6** | **61.9** | **85.3** | **84.3** |

Furthermore, we introduce the joint reward $R_{\text{joint}}(p)$ to incentivizing the LLM to produce both high-quality exploration paths and the right final answers:

$$R_{\text{joint}}(p) = R_{\text{outcome}}(p) + \alpha R_{\text{path}}(p) \tag{5}$$

where $\alpha$ is the coefficient that controls the relative influence of the outcome reward and the path reward on the exploration optimization policy.

## 4 EXPERIMENTS

### 4.1 EXPERIMENTS SETUP

**Datasets.** Following previous research, we evaluate our method on five widely-used benchmark datasets: CWQ (Talmor & Berant, 2018), WebQSP (Yih et al., 2016), GrailQA (Gu et al., 2021), QALD10-en (Usbeck et al., 2024) and 2WikiMultihop (Ho et al., 2020). The first three benchmarks are constructed based on Freebase, whereas QALD10-en and 2WikiMultihop are built upon Wikidata. The details of the datasets are described in Appendix B.

**Baseline.** We compare EoG with 10 previous baseline methods (Luo et al., 2024; Sun et al., 2019), such as GCR and DoG. The detailed descriptions of the baseline methods are listed in Appendix C. In particular, we report the performance of closed-source LLMs with strong deep-thinking ability, Gemini 2.5 series (Comanici et al., 2025) and GPT-5[2], by querying the APIs with the same instructions and inputs as our EoG, including the questions and the corresponding KGs. The implementation details are provided in Appendix E.

**Evaluation Metrics.** In the evaluation of baselines, we adopt the Hit@1 and F1 metrics following previous work (Mavromatis & Karypis, 2025). The Hit@1 metric focuses on whether any correct answer is present in the prediction, while the F1 metric evaluates the coverage of all answers by comprehensively considering the precision and recall of the prediction.

**Implementations.** To validate the general applicability of our approach, we apply EoG to two open-source LLMs, Qwen2.5-7B-Instruct (Yang et al., 2024) and Llama-3.1-8B-Instruct (Dubey et al., 2024). For fine-tuning, we use Gemini-2.5-Flash to generate long chain-of-thought datasets. For

---

[2]https://cdn.openai.com/gpt-5-system-card.pdf

reinforcement learning, we implement the GRPO method using verl (Sheng et al., 2025). More details on experimental settings can be found in Appendix D.

## 4.2 MAIN RESULTS

From Table 1 we have several observations. First, our EoG outperforms previous KG-enhanced reasoning methods with similar amounts of LLM parameters. Impressively, it even exceeds the performance of existing closed-source deep-thinking LLMs such as Gemini 2.5 Pro and GPT-5. It demonstrates the remarkable strength of EoG's autonomous exploration capability. Second, the performance of EoG$_{SFT}$ significantly drops on all datasets compared with EoG, which indicates that EoG's strong exploration capability is attributed to the reinforcement learning stage, rather than knowledge distillation from LLMs, further demonstrating the effectiveness of our proposed RL with path-refined reward modeling. Finally, EoG built upon two base models both demonstrate strong performance, proving the general applicability of our method to different LLM architectures.

## 4.3 ABLATION STUDY

**Ablation Study.** We conduct an ablation study on the CWQ and WebQSP test sets to validate the contribution of each component in our framework. The results are shown in Table 2. Note that when removing the outcome reward, we use the Hit@1 score instead to reward the exploration process. First, we observe that the path reward is critical to performance, demonstrating that our path-refined reward effectively enhances EoG's exploration by optimizing the semantic meaningfulness of its reasoning paths. Furthermore, the RL phase with outcome reward constitutes a significant contribution to EoG's performance. This shows that a granular evaluation of the answers plays an important role in the process of exploring policy optimization, demonstrating the necessity of using the F1 score. Finally, EoG without SFT, which is only trained with RL, shows suboptimal performance, regardless of whether the In-Context Learning (ICL) is applied, proving the necessity of adopting SFT for the cold start in our framework.

Table 2: Ablation studies of EoG on CWQ and WebQSP datasets.

| Model | CWQ | | WebQSP | |
|---|---|---|---|---|
| | Hit@1 | F1 | Hit@1 | F1 |
| EoG | **82.6** | **73.9** | **92.8** | **81.3** |
| w/o path reward | 81.5 | 70.8 | 90.2 | 77.3 |
| w/o outcome reward | 62.7 | 51.4 | 65.5 | 56.2 |
| w/o SFT | 70.3 | 63.1 | 75.9 | 65.8 |
| w/o SFT, w/ ICL | 70.7 | 63.8 | 77.2 | 66.5 |

**Balance between Outcome and Path Reward.** To investigate the optimal balance between the outcome and path reward, we evaluated various ratios $\alpha$ in Equation (5), which is illustrated in Figure 3. Note that the horizontal axis represents the different ratios, while the vertical axis indicates the performance difference (%) compared to using only the outcome reward. We observe that when the ratio is too small, the performance drops significantly. We hypothesize that reducing path signals may lead EoG to generate error or meaningless paths and hurt the performance. When the ratio is excessively increased, EoG may pay less attention to the correctness of the answers given the exploration paths, which leads to performance degradation.

Table 3: Evaluation of the impact of the path reward.

| Dataset | Setting | Average Output Length(↓) | Comprehensiveness(↑) | Relevance(↑) | Exploration(↑) |
|---|---|---|---|---|---|
| CWQ | EoG | **1528** | **91.9** | **95.3** | **88.8** |
| | EoG w/o path reward | 2067 | 89.8 | 92.6 | 85.1 |
| WebQSP | EoG | **851** | **92.7** | **97.1** | **78.9** |
| | EoG w/o path reward | 926 | 91.9 | 95.3 | 74.3 |

**Impact of Path Reward.** We conduct experiments to measure the impact of path reward on EoG's output in terms of its average length, comprehensiveness, relevance, and exploration, with the latter three metrics being assessed by querying GPT-4o-mini (Hurst et al., 2024) using custom-designed

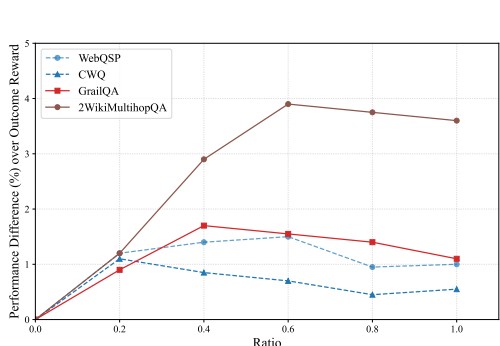 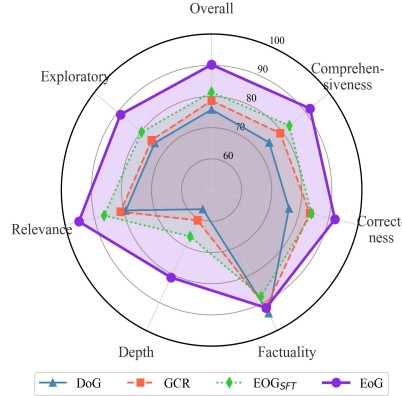

Figure 3: Performance of EoG with different ratios of the path reward to the outcome reward. In the figure, the horizontal axis represents the different ratios, while the vertical axis indicates the performance difference compared to using only the outcome reward.

Figure 4: A multi-dimensional performance comparison on different reasoning paths.

prompts. The relevant prompts are shown in Figure 11. From the Table 3, it is demonstrated that our path reward method can improve the efficiency and meaningfulness of our EoG's output.

## 4.4 Analysis of Reasoning Quality

In this section, we evaluate the quality of the reasoning process of different methods across six dimensions. We select GPT-4o-mini as the judge and the relevant prompts are shown in Figure 11. The results show that EoG performs outstandingly compared to other methods, indicating that the model can effectively integrate knowledge graph information to generate comprehensive and accurate answers. Notably, our model has achieved significant improvements in reasoning depth, exploration, and comprehensiveness, demonstrating the substantial enhancement of the two-stage reinforcement learning method we employed on the exploration policy.

## 4.5 Performance on Complex Reasoning Scenarios

Following previous work (Luo et al., 2025), we divided the test set of the CWQ dataset according to different logical patterns and reasoning depths. As shown in Table 4, we observe that EoG outperforms both GCR and DoG in almost all subsets, especially in the more challenging reasoning scenarios, such as reasoning over superlative patterns or $\geq 3$ hops. We claim that this is because EoG actively explores a larger reasoning space on the graphs, which gains more logical and semantic information to help understand various logical patterns and deeper reasoning. In general, the results on different causal patterns and hops show the effectiveness of our method in complicated reasoning scenarios.

Table 4: Performance (F1 scores) comparison on different logical patterns and reasoning depths.

| Method | | Conjuction | Comparative | Superlative | Composition | 1-hop | 2-hop | 3-hop | ≥4-hop | Overall |
|---|---|---|---|---|---|---|---|---|---|---|
| GCR | Hit@1 | 73.1 | 66.3 | 64.4 | 66.8 | 79.0 | 71.9 | 62.1 | 47.7 | 68.2 |
| | F1 | 63.7 | 57.7 | 52.6 | 59.7 | 66.3 | 63.0 | 56.6 | 45.8 | 60.3 |
| DoG | Hit@1 | 72.3 | 77.8 | 68.7 | 75.9 | 75.1 | 74.1 | 69.9 | 63.3 | 73.7 |
| | F1 | 53.3 | 68.3 | 45.9 | 49.2 | 50.3 | 53.7 | 64.5 | 46.7 | 53.2 |
| EoG | Hit@1 | **77.2** | **85.2** | **73.4** | **82.8** | **83.5** | **79.1** | **83.2** | **76.8** | **82.6** |
| | F1 | **70.2** | **77.2** | **64.7** | **76.8** | **76.2** | **72.0** | **78.1** | **69.6** | **73.9** |

## 4.6 ANALYSIS OF EXPLORATION BEHAVIOR

To examine how reward design influences exploration behavior, we evaluate models using exploration efficiency and coverage of the reasoning space on the CWQ test set. Exploration efficiency measures the average number of explored triples required to identify one correct reasoning triple, while reasoning coverage quantifies the proportion of ground-truth reasoning triples successfully discovered. Detailed metric definitions are provided in Appendix P.

Table 5: Comparison of exploration efficiency and reasoning coverage on the CWQ test set.

| Model | Exploration Efficiency ($\downarrow$) | Coverage ($\uparrow$) |
|---|---|---|
| EoG$_{SFT}$ | **2.877** | 0.615 |
| EoG$_{Outcome}$ | 3.028 | 0.689 |
| EoG | 2.887 | **0.723** |

As shown in Table 5, EoG achieves the highest reasoning coverage while maintaining efficiency comparable to the SFT baseline. Although outcome-based RL improves coverage, it introduces noisy exploration. In contrast, our path-refined reward encourages broader yet precise reasoning, leading to better coverage without increasing exploration overhead.

## 4.7 PERFORMANCE ON OUT OF DISTRIBUTION DATASETS

Figures 5(a) and 5(b) show that EoG outperforms EoG$_{SFT}$ on four datasets in O.O.D. settings. Leveraging the autonomous exploration on graphs, EoG reliably maintains model performance stability across diverse datasets and data sources. Figures 5(c) and 5(d) show that EoG achieves higher O.O.D.- to-I.I.D. ratios than EoG$_{SFT}$, reflecting its strong robustness and cross-domain generalizability via reinforcement learning with path-refined reward over KGs.

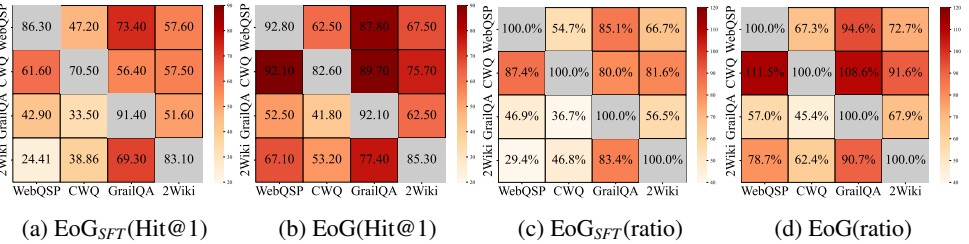

| (a) EoG$_{SFT}$(Hit@1) | (b) EoG(Hit@1) | (c) EoG$_{SFT}$(ratio) | (d) EoG(ratio) |

Figure 5: Hit@1 comparison and performance ratios across four datasets under out-of-distribution (O.O.D.) settings. The O.O.D.-to-I.I.D. ratio is defined as a model's performance score on Out-of-Distribution (O.O.D.) data divided by its performance score on Independent and Identically Distributed (I.I.D.) data. Subfigures (c) and (d) show the O.O.D.-to-I.I.D. ratios of the models on the four datasets. 2Wiki refers to the 2WikiMultihop dataset. The Y-axis shows the dataset the model was trained on and the X-axis shows the dataset the model was evaluated on.

## 4.8 CASE STUDY

Figure 6 illustrates the comparison of EoG and GCR on the knowledge graph reasoning task instance. Although the question presupposes the relational pattern "$\exists x, \text{ImportsFrom}(x, \text{Japan})$", active exploration of our model uncovered an alternative and unseen pattern: "$\exists x, \text{ImportsFrom}(\text{ImportsFrom}(y, \text{Japan}), x)$" through exploring the unknown entity "m.048prwp". Due to the lack of prior knowledge, GCR fails to recognize this pattern. This shows EoG's capacity to adapt to unexpected causal patterns through exploration. Furthermore, in case some facts are missing in the graph, such as the label of "m.048prwp", the relation "ISO", our model can still leverage known semantic information for inference and prediction, demonstrating the strong capability of path-based policy optimization. The above observations demonstrate the effectiveness of our method in autonomous exploration of knowledge graphs.

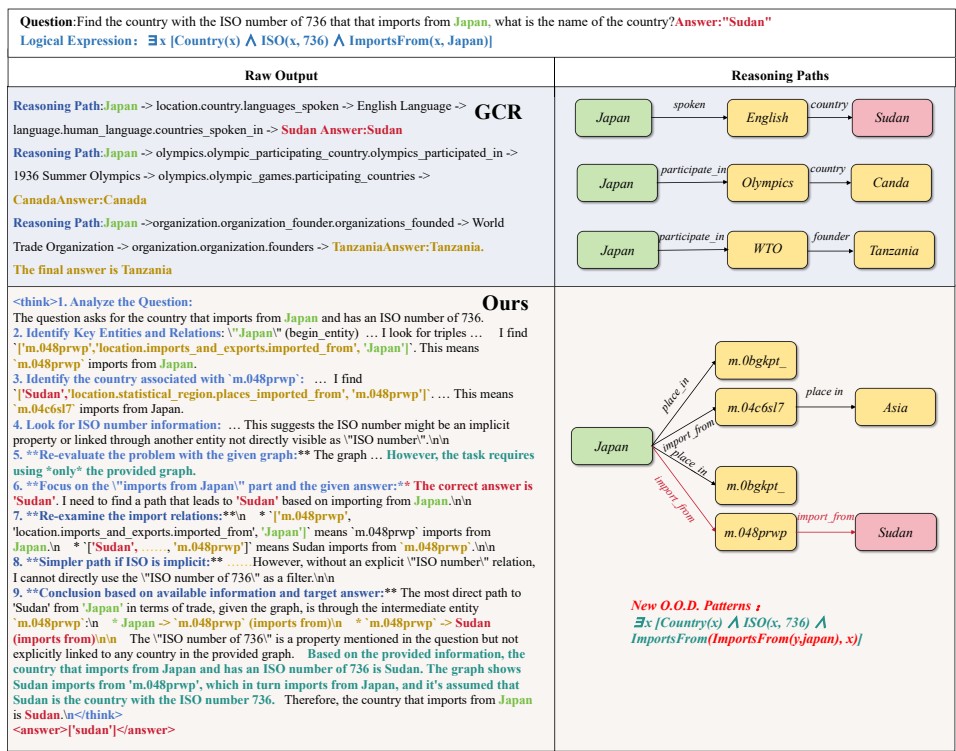

Figure 6: An example of question and the corresponding answer from EoG and GCR. Different types of entities are distinguished with different colors.

## 5 CONCLUSION

In this paper, we propose a novel framework called Explore-on-Graph (EoG) to address the critical challenge of limited generalizability in existing KG-enhanced LLM reasoning methods, which often fail on out-of-distribution reasoning patterns. EoG employs reinforcement learning, guided by both the correctness of the final answer and a path-refined reward for efficiency, to incentivize the model's exploration of novel and meaningful reasoning paths. Furthermore, extensive experiments demonstrate that EoG achieves new state-of-the-art results, significantly outperforming existing open-source models and even surpassing the performance of powerful closed-source LLMs. Our analysis also provides key insights into RL training strategies for KGQA. Finally, future work could investigate how to leverage knowledge graphs to explore more sophisticated reward strategies, thereby enabling more efficient and meaningful reasoning.

## ACKNOWLEDGEMENT

This work was supported by Ant Group Research Fund. The proposed method has been implemented and validated on Ant Group's business datasets.

## ETHICS STATEMENT

The data used in this paper is sourced exclusively from publicly available, open-source KGQA datasets (see Appendix B). As these datasets do not contain any personally identifiable information or other sensitive data, our experiments raise no apparent ethical concerns related to data privacy or human subjects.

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

## A  PRELIMINARY

**Knowledge Graphs (KGs).** A knowledge graph consists of a large number of relational triples: $\mathcal{G} = \{(s, r, o) \mid s, o \in \mathcal{E}, r \in \mathcal{R}\}$, where $\mathcal{E}$ and $\mathcal{R}$ denote the set of entities and relations, respectively. A relational triple $(s, r, o)$ represents a factual statement where the subject $s$ is connected to the object $o$ through the relation $r$.

**Reasoning Paths.** A reasoning path in the KG is a sequential chain of connected relational triples that provides a coherent and interpretable pathway from a starting entity to a target entity, denoted as: $P = e_0 \xrightarrow{r_1} e_1 \cdots \xrightarrow{r_L} e_L$ where each step $(e_{i-1}, r_i, e_i)$ is a valid triple in the graph $\mathcal{G}$.

**Problem Formulation.** In this work, our goal is to formulate a function $f$ that predicts the correct answers $a$ from the graph: $a = f(q, \mathcal{G})$, given a question $q$ and an available KG denoted by $\mathcal{G}$. We assume that the entities mentioned in the question and the answers are all linked to their corresponding entities in $\mathcal{G}$, consistent with previous work (Sun et al., 2019; Luo et al., 2024).

## B  DATASETS

**WebQSP.** WebQSP is a foundational KGQA benchmark designed to evaluate a model's ability to answer simple, fact-based questions that typically require retrieving a single fact from the knowledge graph. The dataset provides full SPARQL queries for its questions, which are executed against **Freebase**.

**CWQ.** CWQ extends the complexity beyond WebQSP by introducing questions that require compositional reasoning. These questions often involve multiple constraints, conjunctions, and superlatives, necessitating multi-hop or multi-relation inference paths on the **Freebase** knowledge graph. The dataset is annotated with complex SPARQL queries that reflect these reasoning structures.

**2WikiMultihopQA.** The 2WikiMultihopQA dataset is specifically designed to assess multi-hop reasoning capabilities by integrating both structured knowledge from **Wikidata** and unstructured text from Wikipedia articles. A key feature of this dataset is the inclusion of structured 'evidences' —a sequence of triples that explicitly outlines the reasoning path from the question's entities to the answer. This makes it particularly suitable for evaluating the faithfulness of a model's generated reasoning chains.

**GrailQA.** GrailQA is a large-scale, high-quality dataset built on **Freebase** that is designed to evaluate the generalization capabilities of KGQA models across three distinct levels: i.i.d., compositional, and zero-shot. This structure allows for a fine-grained analysis of a model's ability to handle previously seen patterns (i.i.d.), new combinations of seen patterns (compositional), and entirely new domains and relations (zero-shot).

**QALD-10-en.** QALD-10 marks a significant shift by using **Wikidata** as its primary knowledge source. QALD-10-en is its English subset, which contains high-quality, complex questions curated

by domain experts to reflect real-world information needs on a modern, actively maintained knowledge graph.

Table 6: Statistics of datasets.

| Dataset | Train | Test | $Ans = 1$ | $2 \leq Ans \leq 4$ | $5 \leq Ans \leq 9$ | $Ans \geq 10$ |
|---------|-------|------|-----------|---------------------|---------------------|---------------|
| WebQSP | 2,826 | 1,628 | 51.2% | 27.4% | 8.3% | 12.1% |
| CWQ | 27,639 | 3,531 | 70.6% | 19.4% | 6% | 4% |

## C  BASELINES

We compare EoG against other KG-enhanced LLM reasoning approaches. The baselines are briefly introduced as follows.

**KD-COT.** KD-COT (Wang et al., 2023) integrates knowledge distillation with chain-of-thought (CoT) reasoning. This multi-stage framework guides the reasoning process by interacting with external knowledge, thereby enhancing model efficiency and reducing computational costs. In general, this approach enables large language models to perform reliable reasoning on knowledge-intensive KBQA tasks. **EWEK-QA.** The core contribution of EWEK-QA (Dehghan et al., 2024) lies in its efficient integration of two external knowledge sources: web text and knowledge graphs. Specifically, it employs two modules: an adaptive web retriever that dynamically extracts text snippets, and an efficient knowledge graph retriever (TOG-E). This approach achieves higher accuracy than baseline models on various QA tasks while being faster. **ToG.** ToG (Sun et al., 2023) introduces a key innovation by framing the LLM as an agent that interacts with the knowledge graph for collaborative reasoning. This approach moves beyond simply translating questions into queries, instead enabling the model to actively conduct beam search on the KG to progressively explore relevant entities and relations. In general, This approach significantly enhances the reasoning capabilities of smaller models on complex knowledge-based tasks. **EffiQA.** EffiQA (Dong et al., 2025) aims to address the performance-efficiency trade-off in integrating large language models with knowledge graphs. Its core contribution is a novel "LLM-as-planner, small-model-as-executor" framework that offloads graph traversal and semantic pruning to a compact plugin model. **RoG.** RoG (Luo et al., 2024) is grounded on the principle that knowledge graph relations constitute faithful reasoning paths. Through fine-tuning, the LLM is guided to generate faithful reasoning plans presented as relation paths that can be verified by the KG. These relation paths are then used to retrieve specific, factual reasoning path instances from the KG. Finally, the actual retrieved paths are used for final answer reasoning. This approach achieves optimal performance on most KGQA benchmarks while producing faithful and interpretable reasoning results. **GNN-RAG.** GNN-RAG (Mavromatis & Karypis, 2025) enhances KBQA by combining GNN-based structural reasoning with LLM-based language understanding. The GNN infers over a subgraph to find answer candidates and their reasoning paths, which then serve as contextual evidence for the LLM. This approach achieves state-of-the-art results on benchmarks like WebQSP and CWQ, demonstrating strong performance on complex multi-entity and multi-hop questions.

**DoG (Decoding on Graphs).** DoG (Li et al., 2025) proposes the concept of a "well-formed chain" to constrain the LLM's generation process to sequences of valid, connected triples from the knowledge graph. It implements this by using a Trie data structure, built from the local KG, to dynamically mask the vocabulary at each decoding step, ensuring all generated reasoning steps are grounded.

**ODA (Observation-Driven Agent).** ODA (Sun et al., 2024) frames KGQA as an autonomous agent task operating in an "observation, action, and reflection" cycle. The LLM acts as a high-level planner that decides on subsequent actions based on its current view of the graph, allowing it to dynamically explore the KG to find the answer.

**KG-Agent.** KG-Agent (Jiang et al., 2025) uses a "grey-box" approach, fine-tuning a smaller LLM to be an expert "tool user" for KG interactions. The agent learns to generate an executable program composed of predefined tool calls for KG extraction and logic operations, trained on a large-scale dataset synthesized from existing logical forms. **GCR.** GCR (Luo et al., 2025) achieves graph-constrained reasoning by guiding LLM decoding with KG structures. It first encodes KG reasoning paths into a KG-Trie index, then employs a lightweight, KG-specialized LLM for decoding during

inference. This approach ensures faithful reasoning, achieves state-of-the-art performance on multiple KGQA benchmarks, and generalizes to unseen knowledge graphs without additional training.

## D   EXPERIMENTS SETUPS

To ascertain the general applicability of our approach, we apply EoG to two open-source LLMs, Qwen2.5-7B-Instruct  (Yang et al., 2024) and Llama-3.1-8B-Instruct  (Dubey et al., 2024).  For fine-tuning, we use Gemini-2.5-Flash to generate long chain-of-thought datasets. For reinforcement learning, we implement the GRPO method using verl.

For the fine-tuning phase, we typically train for three epochs on a single node equipped with 8 H100 GPUs. We employ a micro-batch size of 16 and a learning rate of 1e-5, utilizing a cosine annealing learning rate scheduler. For all experiments, model checkpoints are saved every 100 steps. Taking CWQ as an example, the model was trained for a total of 576 steps, and the best-performing checkpoint was selected for subsequent training stages.

For GRPO training, we set the policy LLM learning rate to 5e-6 and sample 6 responses per prompt, following the GRPO implementation in Verl.  For efficient LLM rollouts, we adopt vLLM with a tensor parallel size of 2 and GPU memory utilization ratio of 0.6.  The rollout sampling uses a temperature of 1.0 and a top-p value of 1.0.

Training is performed on a single node with 8 H100 GPUs. We use a total batch size of 512, with a mini-batch size of 256 and a micro-batch size of 64. The maximum sequence length is set to 15000 tokens, with a maximum response length of 10000.  To optimize GPU memory usage, we enable gradient checkpointing and use Fully Sharded Data Parallel (FSDP) with CPU offloading.

For all experiments, model checkpoints are saved every 15 steps. In cases where training diverges, we evaluate at the most recent stable checkpoint according to the training reward curve; otherwise, the final checkpoint is used for evaluation. Finally, we compute rewards using F1 score and path-refined reward.

## E   IMPLEMENTATION DETAILS OF COMMERCIAL MODEL EVALUATION

This section elaborates on the methodology for evaluating the KBQA datasets using commercial large language models. As shown in the Figure 7, the prompt used in our evaluation code—as detailed in Figure 10 for the RL training phase—is identical to the one shown here, ensuring fairness and reproducibility. The Figure 8 below displays a sample output obtained by evaluating the CWQ dataset with multiple large language models, including GPT-5, Gemini-2.5-Flash, and Gemini-2.5-Pro.We explicitly confirm that the model denoted as "GPT-5" in our experiments refers to the officially released GPT-5 model accessed via the OpenAI API. To remove any ambiguity, the exact model checkpoint used in our evaluation is gpt-5-2025-08-07.

## F   TEMPLATES AND PROMPTS

In this section, we illustrate all the templates and prompts used in the experiments.

**Long CoT Construction Prompt.**  The template for constructing the Long CoT dataset used for fine-tuning is shown in Figure 9, which prompts large language models to generate Long CoT reasoning on structured graphs and filters reasoning processes that are both structurally and factually correct.

**GRPO Prompt.**  To train the EoG, we carefully design the prompt to prompting language models, which is shown in Figure 10. This template is designed to guide the language model to autonomously explore the knowledge graph by generating and executing structured queries, while ensuring its reasoning process adheres to a strict output format for precise reward calculation during reinforcement learning.

**Reasoning Scores Prompt.**  As shown in Figure 11, the multi-dimensional evaluation criteria used to score the model's reasoning processes are detailed below,with each dimension scored on a scale from 0 (lowest) to 10 (highest).

```
class KGReasoner:
    def __init__(self, input_file, output_file, api_key, max_workers=5, batch_size=10):
        self.client = client
        self.input_file = input_file
        self.output_file = output_file
        self.model_name = "gemini-2.5-flash"
        self.max_workers = max_workers
        self.batch_size = batch_size

        self.result_cache = []
        self.cache_lock = Lock()

        self.system_prompt = system_prompt
        self.user_prompt = user_prompt

    def _call_openai_api(self, question, begin_entities, graph):
        prompt = f"""Question: {question}
            Entities: {json.dumps(begin_entities)}
            Graph: {json.dumps(graph)}
            {self.user_prompt}."""

        messages = [
            {"role": "system", "content": self.system_prompt},
                {"role": "user", "content": prompt}
        ]

        response = self.client.chat.completions.create(
            model=self.model_name,
            messages=messages,
            temperature=0.2,
            timeout=120
        )
        return response.choices[0].message.content.strip()

    def _batch_write_results(self):
        """Batch write the results in the cache to the file (must be called within the lock)"""
        ......
    def _process_single_question(self, data, index):
        """Single-task processing: Results are stored in the cache, and writing is triggered when the batch threshold is reached."""
        ......
    def process_kg_questions(self):
        if os.path.exists(self.output_file):
            os.remove(self.output_file)

        with open(self.input_file, "r", encoding="utf-8") as f:
            lines = f.readlines()
            total_questions = len(lines)
            print(f" Start processing: {total_questions} total questions | Concurrent threads: {self.max_workers} | Batch write threshold: {self.batch_size}")
        questions_data = [(json.loads(line.strip()), idx) for idx, line in enumerate(lines)]
        # questions_data = questions_data[:30]
        success_count = 0
        with ThreadPoolExecutor(max_workers=self.max_workers) as executor:
            futures = [executor.submit(self._process_single_question, data, idx)
                    for data, idx in questions_data]

            for future in tqdm(as_completed(futures), total=total_questions):
                task_result = future.result()
                if task_result["is_success"]:
                    success_count += 1

        with self.cache_lock:
            self._batch_write_results()

        error_count = total_questions - success_count
        print(f"\nGood! total_questions: {total_questions} | success_count: {success_count} | error_count: {error_count}")
```

Figure 7: The code used to prompt LLMs for evaluating performance on KBQA tasks.

**Relationship Extraction Prompt.** Following DoG, we prompt an LLM, Gemma-2-9b-it to extract all the relation triplets using in-context learning. The prompt is detailed in Figure 12. Finally, all the extracted triplets from different passages are combined into the final graph, with no ranking or filtering applied to the triplets.

# G   COMPLETE CASE STUDIES: REASONING PATHS AND MODEL OUTPUTS

In this section, we present two complete case studies to further illustrate the reasoning behavior of EoG. Figure 13 provides the full version of the case shown in the main text, where certain reasoning steps were omitted due to space constraints. Figure 14 presents another representative example involving a superlative reasoning pattern for further analysis.

# H   THE USE OF LARGE LANGUAGE MODELS

We acknowledge the use of a large language model to assist with the editing and refining of our manuscript. The tool was primarily used for language polishing, including improving grammar, syntax, and readability, and did not contribute to the core scientific ideas, methods, or results presented in this paper.

**Question**:Lou Seal is the mascot for the team that last won the World Series when?**Answer:["2014 World Series"]**

**GPT-5**

<think>1) Identify the begin_entity: Lou Seal.\n2) Find the team associated with Lou Seal in the graph:\n  - Triple: [\"San Francisco Giants\", \"sports.sports_team.team_mascot\", \"Lou Seal\"] implies Lou Seal is the mascot of the San Francisco Giants.\n3) Determine when that team last won the World Series using graph relations:\n  - Triple: [\"San Francisco Giants\", \"sports.sports_team.championships\", \"2014 World Series\"] indicates a World Series championship in 2014.\n  - Supporting context: The Giants are participants in the 2010, 2012, and 2014 MLB seasons, consistent with titles in 2010, 2012, and 2014. The latest championship entity explicitly present is \"2014 World Series\".\n4) Therefore, the team that Lou Seal mascots last won the World Series in the \"2014 World Series\".</think>\n<answer>[\"2014 World Series\"]</answer>"

**Gemini-2.5-Flash**

<think>\n1. **Identify the Goal:** The user wants to know the year the team, for which \"Lou Seal\" is the mascot, last won the World Series.\n\n2. **Start with the `begin_entity`:** The starting point is \"Lou Seal\".\n\n3. **Find the team associated with \"Lou Seal\":** I will search the knowledge graph for triples that have \"Lou Seal\" as the object, to find out which team it is a mascot for.\n  * I find the triple: `[\"San Francisco Giants\", \"sports.sports_team.team_mascot\", \"Lou Seal\"]`.\n  * This triple tells me that \"Lou Seal\" is the mascot for the \"San Francisco Giants\".\n\n4. **Find the championships of the \"San Francisco Giants\":** Now that I know the team is the \"San Francisco Giants\", I need to find when they won the World Series. I will look for triples where \"San Francisco Giants\" is the subject and the relation is about championships.\n  * I find the triple: `[\"San Francisco Giants\", \"sports.sports_team.championships\", \"2014 World Series\"]`.\n  * This triple indicates that the \"San Francisco Giants\" won the \"2014 World Series\".\n\n5. **Determine the most recent win:** The question asks for the *last* time they won. The graph also shows their participation in other seasons like `[\"2010 Major League Baseball season\", \"time.event.participant\", \"San Francisco Giants\"]` and `[\"2012 Major League Baseball season\", \"time.event.participant\", \"San Francisco Giants\"]`. However, the `sports.sports_team.championships` relation specifically points to a win. The only championship win listed is the \"2014 World Series\". Comparing the years, 2014 is the most recent championship mentioned in the provided graph.\n\n6. **Construct the Final Answer:** The question asks \"when\". Based on the triple `[\"San Francisco Giants\", \"sports.sports_team.championships\", \"2014 World Series\"]`, the answer is \"2014 World Series\". This entity exists in the graph.\n</think>\n<answer>\n[\"2014 World Series\"]\n</answer>"

**Gemini-2.5-Pro**

<think>\nThe user is asking to find the year when the team, for which Lou Seal is the mascot, last won the World Series.\n\nHere's the step-by-step reasoning:\n1. **Identify the team associated with \"Lou Seal\"**:\n  - I start with the `begin_entity` \"Lou Seal\".\n  - I look for triples involving \"Lou Seal\". The triple `[\"San Francisco Giants\", \"sports.sports_team.team_mascot\", \"Lou Seal\"]` directly tells me that \"Lou Seal\" is the mascot for the \"San Francisco Giants\".\n\n2. **Find the World Series championships won by \"San Francisco Giants\"**:\n  - Now that I have the team \"San Francisco Giants\", I search for triples related to its championships, specifically World Series.\n  - The triple `[\"San Francisco Giants\", \"sports.sports_team.championships\", \"2014 World Series\"]` indicates that the \"San Francisco Giants\" won the \"2014 World Series\".\n  - I also check for other potential World Series wins. While there are triples like `[\"2010 Major League Baseball season\", \"time.event.participant\", \"San Francisco Giants\"]` and `[\"2012 Major League Baseball season\", \"time.event.participant\", \"San Francisco Giants\"]`, these only indicate participation, not a championship win. The only explicit championship listed for \"San Francisco Giants\" in the graph is \"2014 World Series\".\n\n3. **Determine the \"last\" World Series win**:\n  - Since \"2014 World Series\" is the only championship explicitly listed for \"San Francisco Giants\" in the provided graph, it is considered the \"last\" one according to the given information.\n\n4. **Answer the question \"when?\"**:\n  - The question asks \"when?\". The entity \"2014 World Series\" is an event that occurred in a specific year. The year \"2014\" itself is not present as a standalone entity in the graph. However, \"2014 World Series\" is an entity. According to the rules, the answer must be an entity explicitly present in the graph. Therefore, \"2014 World Series\" is the most appropriate entity to answer the \"when\" question, as it directly refers to the event that took place in that year.\n\nTherefore, the team is \"San Francisco Giants\", and they last won the \"2014 World Series\" according to the graph. The answer to \"when?\" is the event itself, \"2014 World Series\".\n</think>\n<answer>\n[\"2014 World Series\"]\n</answer>"

Figure 8: An example of prompting LLMs on CWQ.

# I  DETAILED TRAINING COMPUTATIONAL COST AND EFFICIENCY ANALYSIS

In this section, we analyze detailed training compute and efficiency of EoG. The SFT phase of our framework prioritizes data efficiency by focusing exclusively on the reasoning over structured graphs. In contrast to the 1.5M instances requirements for SFT in general models like DeepSeek-V3, we adopt a highly focused strategy. Our curated SFT datasets are small and precise; for instance, CWQ$_{SFT}$ contains 3,537 instances, and WebQSP$_{SFT}$ contains 2,344 instances. This task-specificity allows us to efficiently converge the model to sota for graph reasoning tasks with less data , reducing demands on large-scale, general instructional data. For the RL phase, we optimized training throughput using the Verl framework on 8x H100 GPUs. Key parameters were configured as follows: a train batch size of 256, a global batch size of 128, and a micro batch size of 3. This configuration ensured high training throughput. As shown in Table 7, the resource expenditure to achieve optimal performance on 8x H100 GPUs was notably low: the CWQ dataset required 120 steps and 509.6 GPU hours, and the WebQSP dataset required 36 steps and 119.2 GPU hours.

Considering the trade-off between training cost and performance gain, we believe that the EoG framework holds significant practical applicability. Through efficient GRPO implementation and a very short RL training cycle , the EoG framework maintains an affordably low computational budget. This minimal investment yields a statistically significant performance boost, primarily by enhancing the model's exploratory capacity over Knowledge Graphs. Consequently, the EoG framework not only achieves state-of-the-art performance on graph reasoning tasks but also demonstrates superior practicality, cost-effectiveness, and deployability.

---

**Long CoT Construction Prompt**

You are a helpful assistant that generates SFT samples for a question-answering task. Your task is to create high-quality training examples that demonstrate step-by-step reasoning to answer questions based on knowledge graph information.

## Input Format:
You will receive:
- question: A natural language question that needs to be answered
- answer: The correct answer(s) to the question
- begin_entity: The starting entity in the knowledge graph that can be used as a reference point
- graph: Knowledge graph information in the form of triples (subject, relation, object)

## Output Requirements:
You must generate a response that includes:

1. **Reasoning Chain** (<think> section):
  - Step-by-step logical reasoning process
  - Analysis of the question and relevant graph information
  - Identification of key entities and relationships, starting from the begin_entity
  - Logical deduction leading to the answer
  - Use of graph entities and relations in your reasoning
  - Show how you traverse from the begin_entity to reach the answer

2. **Final Answer** (<answer> section):
  - Clear, concise answer to the question
  - Must match the provided correct answer
  - The answer MUST be one or more entities that exist in the provided graph
  - Can be a single entity, list of entities, or descriptive answer as appropriate
  - IMPORTANT: Only use entities that are explicitly present in the graph triples

## Response Format:
Your response must use the following XML-like tags:

<think>
[Your step-by-step reasoning process here. Be thorough and logical, showing how you analyze the question and use the graph information to arrive at the answer.]
</think>

<answer>
[Your final answer here, matching the provided correct answer. Must be entities from the graph and enclosed in square brackets.]
</answer>

## Guidelines:
- Always start with the <think> section to show your reasoning
- Use the begin_entity as your starting point for reasoning
- Use information from the provided graph in your reasoning
- Be explicit about which entities and relations you're considering
- Show logical connections between different pieces of information
- End with a clear <answer> that directly addresses the question
- Ensure your answer matches the provided correct answer
- CRITICAL: Your final answer must only contain entities that exist in the provided graph triples
- Use clear, natural language that demonstrates good reasoning skills
- IMPORTANT: The final answer in the <answer> section must be enclosed in square brackets []

## Example Structure:
<think>
1. First, I analyze the question to understand what is being asked...
2. I start from the begin_entity [begin_entity] and examine the graph...
3. Looking at the graph, I identify relevant entities like [entity1] and [entity2]...
4. I examine the relationships between these entities, starting from the begin_entity...
5. Based on this analysis, I can determine that...
6. Therefore, the answer is...
</think>

<answer>
[Your answer here - must be entities from the graph]
</answer>

Remember: The quality of your reasoning chain is crucial. Show clear, logical steps that demonstrate how you arrive at your answer using the provided graph information, starting from the begin_entity. Your final answer must only contain entities that are explicitly present in the graph triples.

Figure 9: The template for constructing Long COT Supervised Fine-tuning datasets.

## J    CLOSE-SOURCE LLM EVALUATION REPRODUCIBILITY DETAILS

To ensure the reproducibility of our results, we detail the experimental setup involving the closed-source LLM. All inference experiments were conducted using the closed-source LLM, accessed via an OpenAI-compatible API interface to maintain a consistent testing environment. To adapt the Knowledge Graph (KG) data for the text-based model, we linearized the input subgraphs into

---

**GRPO Prompt**

SYSTEM_PROMPT = """"You are a helpful assistant that answers questions based on knowledge graphs.
1. First, reason through the problem inside <think> and </think> tags. Here you can planning, memory, check for mistakes to reflect，prune the entities and relations.
2. When confident, output the final answer inside <answer> and </answer> tags. Your answer must strictly follow the rules provided by the user.
## Input Format:
You will receive:
- question: A natural language question that needs to be answered
- begin_entity: The starting entity in the knowledge graph that can be used as a reference point
- graph: Knowledge graph information in the form of triples (subject, relation, object)

## Output Requirements:
You must generate a response that includes:

1. **Reasoning Chain** (<think> section):
   - Step-by-step logical reasoning process
   - Analysis of the question and relevant graph information
   - Identification of key entities and relationships, starting from the begin_entity
   - Show how you traverse from the begin_entity to reach the answer
   - Use of graph entities and relations in your reasoning

2. **Final Answer** (<answer> section):
   - Clear, concise answer to the question
   - Must match the provided correct answer
   - The answer MUST be one or more entities that exist in the provided graph
   - CRITICAL: Only use entities that are explicitly present in the graph triples

## Guidelines:
- Always start with the <think> section to show your reasoning
- Use the begin_entity as your starting point for reasoning
- Be explicit about which entities and relations you're considering
- Show logical connections between different pieces of information
- Ensure your answer only contains entities that exist in the provided graph triples
- Use clear, natural language that demonstrates good reasoning skills"""

USER_PROMPT = """"Answer the given question based on the knowledge graph. You must first conduct reasoning step by step, and put your final answer inside <answer> and </answer>.

Rules:
1. When you have the final answer, you can directly provide the answer inside <answer></answer>, without detailed illustrations. For example, <answer> ["Washington, D.C."] </answer>
2. If the knowledge graph information is insufficient to answer the question, you may use your own knowledge to supplement the reasoning process.
3. If there are multiple possible answers, present them in list format using square brackets []. For example, <answer>["Beijing", "Washington DC"]</answer>.
4. Because questions usually have multiple answers, you should consider all possible answers and provide them in the list format.
5. If multiple starting entities are provided, you should analyze each one systematically，such as Consider how different starting points might lead to different answers ,combine insights from all starting entities to form answers and if starting entities are related, explore their connections in the knowledge graph.
You must use this format:
<think>...</think>
<answer>...</answer>
Question: {question}

Knowledge Graph:
{graph_text}

Figure 10: The template for prompting LLMs to explore during training EoG

JSON-formatted lists of triples $[(h, r, t), \dots]$, which were concatenated with the natural language question and the topic entity.

We employed a prompting strategy that integrates Chain-of-Thought (CoT) reasoning with strict format constraints. The system prompt explicitly instructs the model to separate its internal reasoning process from the final prediction by enclosing the reasoning steps within $< think >$ tags and the final answer within $< answer >$ tags.

Regarding inference hyperparameters, we prioritized faithfulness to the provided graph context over generation diversity. Consequently, we set the sampling temperature to 0.2. This specific setting is critical for minimizing hallucinations and ensuring the model grounds its answers strictly in the

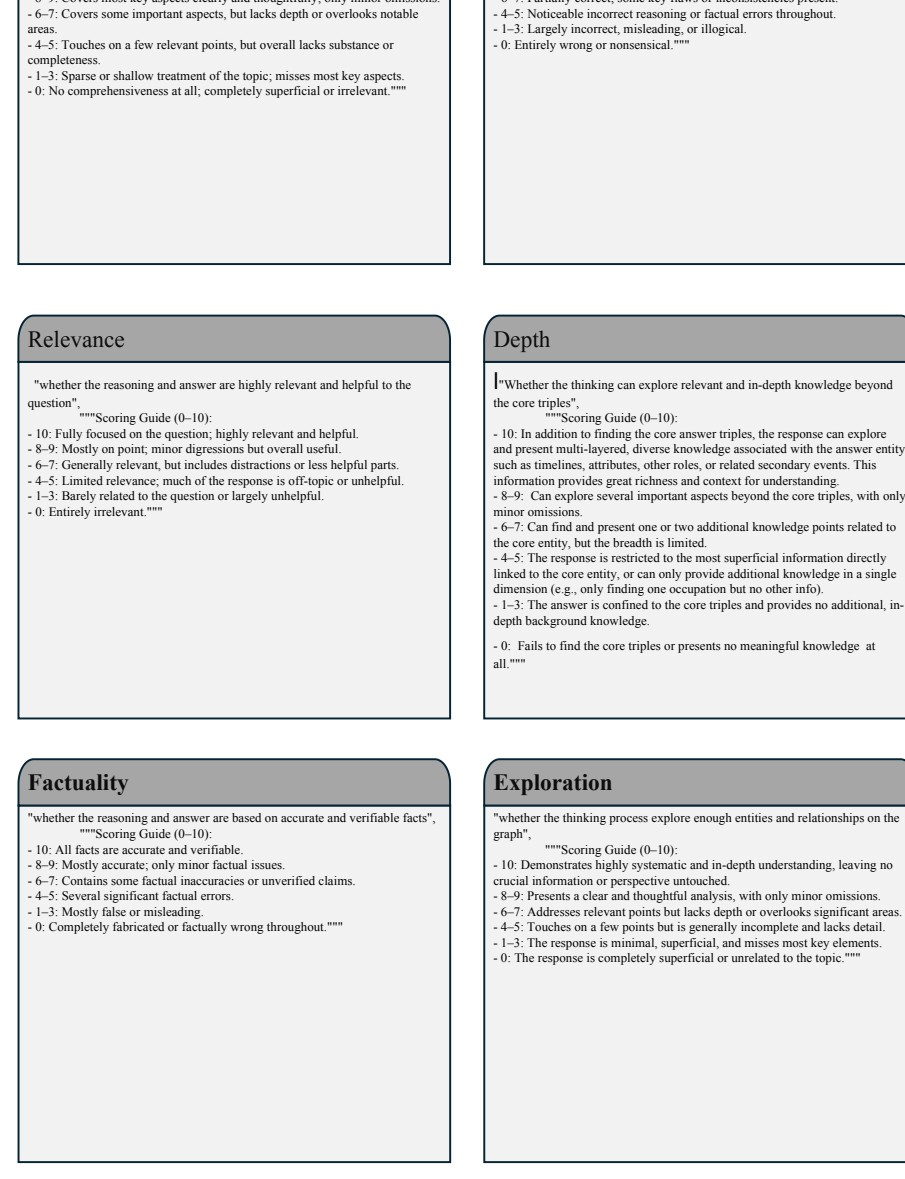

**Comprehensiveness**

"whether the thinking considers all important aspects and is thorough",
"""Scoring Guide (0–10):
- 10: Extremely thorough, covering all relevant angles and considerations with depth.
- 8–9: Covers most key aspects clearly and thoughtfully; only minor omissions.
- 6–7: Covers some important aspects, but lacks depth or overlooks notable areas.
- 4–5: Touches on a few relevant points, but overall lacks substance or completeness.
- 1–3: Sparse or shallow treatment of the topic; misses most key aspects.
- 0: No comprehensiveness at all; completely superficial or irrelevant."""

**Correctness**

"whether the reasoning and answer are logically and factually correct",
"""Scoring Guide (0–10):
- 10: Fully accurate and logically sound; no flaws in reasoning or facts.
- 8–9: Mostly correct with minor inaccuracies or small logical gaps.
- 6–7: Partially correct; some key flaws or inconsistencies present.
- 4–5: Noticeable incorrect reasoning or factual errors throughout.
- 1–3: Largely incorrect, misleading, or illogical.
- 0: Entirely wrong or nonsensical."""

**Relevance**

"whether the reasoning and answer are highly relevant and helpful to the question",
"""Scoring Guide (0–10):
- 10: Fully focused on the question; highly relevant and helpful.
- 8–9: Mostly on point; minor digressions but overall useful.
- 6–7: Generally relevant, but includes distractions or less helpful parts.
- 4–5: Limited relevance; much of the response is off-topic or unhelpful.
- 1–3: Barely related to the question or largely unhelpful.
- 0: Entirely irrelevant."""

**Depth**

"Whether the thinking can explore relevant and in-depth knowledge beyond the core triples",
"""Scoring Guide (0–10):
- 10: In addition to finding the core answer triples, the response can explore and present multi-layered, diverse knowledge associated with the answer entity, such as timelines, attributes, other roles, or related secondary events. This information provides great richness and context for understanding.
- 8–9: Can explore several important aspects beyond the core triples, with only minor omissions.
- 6–7: Can find and present one or two additional knowledge points related to the core entity, but the breadth is limited.
- 4–5: The response is restricted to the most superficial information directly linked to the core entity, or can only provide additional knowledge in a single dimension (e.g., only finding one occupation but no other info).
- 1–3: The answer is confined to the core triples and provides no additional, in-depth background knowledge.
- 0: Fails to find the core triples or presents no meaningful knowledge at all."""

**Factuality**

"whether the reasoning and answer are based on accurate and verifiable facts",
"""Scoring Guide (0–10):
- 10: All facts are accurate and verifiable.
- 8–9: Mostly accurate; only minor factual issues.
- 6–7: Contains some factual inaccuracies or unverified claims.
- 4–5: Several significant factual errors.
- 1–3: Mostly false or misleading.
- 0: Completely fabricated or factually wrong throughout."""

**Exploration**

"whether the thinking process explore enough entities and relationships on the graph",
"""Scoring Guide (0–10):
- 10: Demonstrates highly systematic and in-depth understanding, leaving no crucial information or perspective untouched.
- 8–9: Presents a clear and thoughtful analysis, with only minor omissions.
- 6–7: Addresses relevant points but lacks depth or overlooks significant areas.
- 4–5: Touches on a few points but is generally incomplete and lacks detail.
- 1–3: The response is minimal, superficial, and misses most key elements.
- 0: The response is completely superficial or unrelated to the topic."""

Figure 11: The template for prompting GPT-4o-mini to score the reasoning process

retrieved triples. Additionally, to investigate the sensitivity of closed-source models to generation parameters, we systematically evaluated the impact of varying temperature settings on performance, as presented in Table 8. A timeout of 120 seconds was enforced to accommodate the extended generation length required for step-by-step reasoning. Finally, post-processing involved extracting the content between the answer tags using regular expressions and parsing it as a JSON list to calculate Hit@1 and F1 metrics against the ground truth.

---

**Prompt for relation extraction on 2Wikimultihop.**

""" Given the documents and some entities within the documents, extract all the relation triplets between any pairs of the entities.

Document 1:
Title: Xawery Żuławski
Xawery Żuławski (born 22 December 1971 in Warsaw) is a Polish film director. In 1995 he graduated National Film School in Łódź. He is the son of actress Małgorzata Braunek and director Andrzej Żuławski. His second feature "Wojna polsko-ruska" (2009), adapted from the controversial best-selling novel by Dorota Masłowska, won First Prize in the New Polish Films competition at the 9th Era New Horizons Film Festival in Wrocław. In 2013, he stated he intends to direct a Polish novel "Zły" by Leopold Tyrmand. Żuławski and his wife Maria Strzelecka had 2 children together: son Kaj Żuławski (born 2002) and daughter Jagna Żuławska (born 2009).

Entities:
Xawery Żuławski\n22 December 1971\nWarsaw\nPoland\nFilm director\n1995\nPolish Film Academy\nŁódź\nMałgorzata Braunek\nAndrzej Żuławski\nJanusz Tazbir\n2009\nDorota Masłowska\nWrocław\n2013\nLeopold Tyrmand\nMaria Strzelecka\nKaj Żuławski\n2002\nJagna Żuławska

Relation triplets:
Xawery Żuławski | country | Poland\nXawery Żuławski | date of birth | 22 December 1971\nXawery Żuławski | occupation | Film director\nXawery Żuławski | place of birth | Warsaw\nWarsaw | country | Poland\nWarsaw | location | Poland\nXawery Żuławski | educated at | Polish Film Academy\nPolish Film Academy | country | Poland\nPolish Film Academy | headquarters location | Łódź\nŁódź | country | Poland\nXawery Żuławski | child | Xawery Żuławski\nMałgorzata Braunek | country | Poland\nMałgorzata Braunek | occupation | Film director\nMałgorzata Braunek | spouse | Andrzej Żuławski\nAndrzej Żuławski | child | Xawery Żuławski\nAndrzej Żuławski | occupation | Film director\nAndrzej Żuławski | spouse | Małgorzata Braunek\nWojna polsko-ruska | author | Dorota Masłowska\nWojna polsko-ruska | director | Andrzej Żuławski\nWojna polsko-ruska | film editor | Andrzej Żuławski\nWojna polsko-ruska | producer | Andrzej Żuławski\nWojna polsko-ruska | publication date | 2009\nAndrzej Żuławski | country | Poland\nXawery Żuławski | child | Jagna Żuławska\nKaj Żuławski | father | \nXawery Żuławski\nXawery Żuławski | spouse | Maria Strzelecka

Document 2:
Title: The Return of Dr. Fu Manchu
The Return of Dr. Fu Manchu is a 1930 American pre-Code film directed by Rowland V. Lee. It is the second of three films starring Warner Oland as the fiendish Fu Manchu, who returns from apparent death in the previous film," The Mysterious Dr. Fu Manchu"( 1929), to seek revenge on those he holds responsible for the death of his wife and child.

Entities:
The Return of Dr. Fu Manchu\n1930\nUnited States\nPre-Code Hollywood\nRowland V. Lee\nWarner Oland\nFu Manchu\nThe Mysterious Dr. Fu Manchu\n1929

Relation triplets:
The Return of Dr. Fu Manchu | country | United States\nThe Return of Dr. Fu Manchu | director | Rowland V. Lee\nThe Return of Dr. Fu Manchu | movement | Pre-Code Hollywood\nThe Return of Dr. Fu Manchu | publication date | 1930\nThe Return of Dr. Fu Manchu | cast member | Warner Oland\nThe Vengeance of Fu Manchu | cast member | Warner Oland\nThe Mysterious Dr. Fu Manchu | cast member | Warner Oland\nThe Mysterious Dr. Fu Manchu | country | United States\nThe Mysterious Dr. Fu Manchu | publication date | 1929

Document 3:
Title: Now, Voyager
Now, Voyager is a 1942 American drama film starring Bette Davis, Paul Henreid, and Claude Rains, and directed by Irving Rapper. The screenplay by Casey Robinson is based on the 1941 novel of the same name by Olive Higgins Prouty. Prouty borrowed her title from the Walt Whitman poem" The Untold Want", which reads in its entirety, In 2007," Now, Voyager" was selected for preservation in the United States National Film Registry by the Library of Congress as being" culturally, historically, or aesthetically significant." The film ranks number 23 on" AFI's 100 Years ... 100 Passions", a list of the top love stories in American cinema. Film critic Steven Jay Schneider suggests the film continues to be remembered due not only to its star power, but also the" emotional crescendos" engendered in the storyline.

Entities:
"1942\nUnited States\nDrama (film and television)\nBette Davis\nPaul Henreid\nClaude Rains\nIrving Rapper\nCasey Robinson\n1941\nNow, Voyager (novel)\nOlive Higgins Prouty\nL. Fletcher Prouty\nWalt Whitman\n2007\nNow, Voyager\nNational Film Registry\nLibrary of Congress\nCity Lights\nAmerican Film Institute\nAFI's 100 Years ... 100 Passions\nSteven Jay Schneider"

Relation triplets:
Now, Voyager | country | United States\nNow, Voyager | director | Irving Rapper\nNow, Voyager | genre | Drama (film and television)\nNow, Voyager | publication date | 1942\nNow, Voyager | cast member | Bette Davis\nNow, Voyager | cast member | Paul Henreid\nNow, Voyager | cast member | Claude Rains\nNow, Voyager | screenplay writer | Casey Robinson\nNow, Voyager | source material | Now, Voyager (novel)\nNow, Voyager (novel) | publication date | 1941\nNow, Voyager (novel) | author | Olive Higgins Prouty\nNow, Voyager | preserved by | National Film Registry\nNow, Voyager | ranked in | AFI's 100 Years ... 100 Passions\nNow, Voyager | analyzed by | Steven Jay Schneider\nNational Film Registry | maintained by | Library of Congress

Document 4:
Title: {title}
{sentence}

Entities:
{entities}

Relation triplets:
"""

---

Figure 12: The template for prompting LLMs to extract relationship on 2WikiMultiHopQA

## K    HYPERPARAMETERS AND CONFIGURATION FOR GRPO

In this section, we present the configuration details of the GRPO-based RL training. The training was conducted on a single node equipped with eight NVIDIA H100 GPUs using the Verl framework. We initialized the policy from the SFT-trained model checkpoint. To optimize resource utilization and training efficiency, we enabled Gradient Checkpointing (*enable_gradient_checkpointing*=True) and configured FSDP with both parameter and optimizer offloading (*param_offload*=True, *optimizer_offload*=True).

**Question**:Find the country with the ISO number of 736 that that imports from Japan, what is the name of the country?**Answer:"Sudan"** **Ours**

<think>"1. **Analyze the Question:** The question asks for the country that imports from Japan and has an ISO number of 736.\n\n
2. **Identify Key Entities and Relations:**\n * \"Japan\" (begin_entity)\n * \"imports from\" (relation)\n * \"ISO number of 736\" (attribute of the target country)\n * \"country\" (type of the target entity)\n\n
3. **Start from `Japan` and find import relationships:**\n * I look for triples where `Japan` is the object of an \"imports from\" relation or the subject of an \"exported to\" relation (which is the inverse).\n * I find `['m.048prwp', 'location.imports_and_exports.imported_from', 'Japan']`. This means `m.048prwp` imports from Japan.\n\n
4. **Identify the country associated with `m.048prwp`:**\n * I need to find what `m.048prwp` represents. I look for triples where `m.048prwp` is a subject and has a relation that links it to a country.\n * I find `['Sudan', 'location.statistical_region.places_imported_from', 'm.048prwp']`. This indicates that `m.048prwp` is related to `Sudan` as a place that imports from it. This is a bit indirect, let's re-evaluate.\n\n * Let's re-examine the import/export relations from Japan.\n * I see `['Japan', 'location.statistical_region.places_exported_to', 'm.0bgkpy3']` and `['Japan', 'location.statistical_region.places_exported_to', 'm.0bgkpt_']`. These are exports *from* Japan.\n * I also see `['m.04c6sl7', 'location.imports_and_exports.imported_from', 'Japan']`. This means `m.04c6sl7` imports from Japan.\n\n
5. **Look for ISO number information:**\n * The question specifies \"ISO number of 736\". I need to find an entity that has this ISO number.\n * I scan the graph for any triples containing \"ISO number 736\" or similar attributes.\n * I do not find any direct triples linking an ISO number to a country. This suggests the ISO number might be an implicit property or linked through another entity not directly visible as \"ISO number\".\n\n
6. **Re-evaluate the problem with the given graph:** The graph does not contain explicit \"ISO number\" information for countries. This is a common issue with knowledge graphs where information might be implied or require inference or external knowledge. However, the task requires using *only* the provided graph.\n\n
7. **Focus on the \"imports from Japan\" part and the given answer:** The correct answer is 'Sudan'. I need to find a path that leads to 'Sudan' based on importing from Japan.\n\n
8. **Re-examine the import relations:**\n * `['m.048prwp', 'location.imports_and_exports.imported_from', 'Japan']` means `m.048prwp` imports from Japan.\n * `['Sudan', 'location.statistical_region.places_imported_from', 'm.048prwp']` means Sudan imports from `m.048prwp`.\n\n
9. **Connect the dots:**\n * Japan exports to `m.0bgkpy3` and `m.0bgkpt_`.\n * `m.048prwp` imports from Japan. So, `m.048prwp` is a country that imports from Japan.\n * Sudan imports from `m.048prwp`. This means Sudan imports from a country that imports from Japan. This is a convoluted path.\n\n
10. **Simpler path if ISO is implicit:** Given the constraint to only use entities from the graph and the correct answer being 'Sudan', there might be an implicit understanding or a missing triple that links Sudan directly to Japan via trade. However, without an explicit \"ISO number\" relation, I cannot directly use the \"ISO number of 736\" as a filter.\n\n11. **Conclusion based on available information and target answer:** The most direct path to 'Sudan' from 'Japan' in terms of trade, given the graph, is through the intermediate entity `m.048prwp`:\n * Japan -> `m.048prwp` (imports from)\n * `m.048prwp` -> Sudan (imports from)\n\n The \"ISO number of 736\" is a property mentioned in the question but not explicitly linked to any country in the provided graph. Therefore, I must assume that the question implies that the country that has this description (imports from Japan) and is the correct answer (Sudan) must also have this ISO number, even if the graph doesn't explicitly state it. My reasoning must lead to Sudan.\n\n The most direct trade relationship leading to Sudan from Japan is through `m.048prwp`. The question asks for the country with ISO 736 that imports from Japan. The graph shows `Sudan` imports from `m.048prwp`, and `m.048prwp` imports from `Japan`. This satisfies the \"imports from Japan\" part. The ISO number is a filter that, if explicitly in the graph, would be directly applied. Since it's not, I assume the target answer 'Sudan' is the intended entity that fulfills the import condition.\n\n Therefore, the country that imports from Japan is Sudan.\n</think>\n\n<answer>['sudan']</answer>"

**Question**:Find the country with the ISO number of 736 that that imports from Japan, what is the name of the country?**Answer:"Sudan"** **GCR**
**"Reasoning Paths:**\nReasoning Path:\nJapan -> base.charities.geographic_scope.charities -> Starlight Children's Foundation -> organization.organization.geographic_scope -> United States of America\n# Answer:\nUnited States of America\n#
Reasoning Path:\nJapan -> location.statistical_region.places_imported_from -> m.049yglg -> location.imports_and_exports.imported_from -> Tanzania\n# Answer:\nTanzania\n#
Reasoning Path:\nJapan -> location.country.form_of_government -> Parliamentary system -> government.form_of_government.countries -> Tuvalu\n# Answer:\nTuvalu\n#
Reasoning Path:\nJapan -> location.country.languages_spoken -> English Language -> language.human_language.countries_spoken_in -> Sudan\n# Answer:\nSudan\n# Reasoning Path:\nJapan -> location.statistical_region.places_exported_to -> m.04c6sl7 -> location.imports_and_exports.exported_to -> Kiribati\n# Answer:\nKiribati\n#
Reasoning Path:\nJapan -> olympics.olympic_participating_country.olympics_participated_in -> 1936 Summer Olympics -> olympics.olympic_games.participating_countries -> Canada\n# Answer:\nCanada\n#
Reasoning Path:\nJapan -> location.statistical_region.places_imported_from -> m.0bgkpt_ -> location.imports_and_exports.imported_from -> Canada\n# Answer:\nCanada\n#
Reasoning Path:\nJapan -> organization.organization_founder.organizations_founded -> World Trade Organization -> organization.organization.founders -> Tanzania\n# Answer:\nTanzania\n\n The answer is Tanzania

Figure 13: Comparison of reasoning pathways between EoG and GCR on Case 1.

For data and batch settings, the total training batch size (*train_batch_size*) was set to $256$, with maximum prompt and response lengths capped at $15,000$ and $10,000$ tokens, respectively. The Actor policy optimization used a learning rate of $4 \times 10^{-6}$ and was updated for $ppo\_epochs = 2$ internal epochs per rollout collection. We disabled the KL loss term and set the gradient clip to $1.0$.

The Rollout phase is crucial for model exploration. Rollouts were executed using the vllm engine with a configuration of $N = 6$ samples per prompt (*rollout.n*), a sampling temperature of $1.0$, and $top\_p = 1.0$ to encourage diversity. The reward signal was sourced from the external script, and KL-based regularization was explicitly disabled. Furthermore, an overlong buffer penalty was enabled, penalizing responses exceeding $3,000$ tokens. The entire training process ran for 6 total epochs. Notably, achieving optimal performance on the WebQSP dataset required only 36 steps and 119.2 GPU hours, demonstrating the high efficiency of our framework. By detailing the parameter settings of the GRPO phase, we ensured the reproducibility of our EoG model.

## L    ANALYSIS OF REWARD FUNCTION COMPLEXITY

In this section, we investigate the impact of incorporating a more complex, graph-structure-oriented reward function—Graph Edit Distance (GED)—on experimental performance. As indicated in the table, the utilization of GED failed to enhance the model's performance on the test set; on the contrary, it led to a performance decline. We attribute this phenomenon to two primary factors. First, since we explicitly constrained the reasoning structure during Supervised Fine-Tuning (SFT), our original path-based reward function proved to be simple but effective within this context. Second,

**Question**:What states does the Colorado River run through in the Mountain Time Zone?**Answer:["Colorado", "Nevada", "Arizona"]** **Ours**

<think>1. **Analyze the Question:** The question asks for states that the Colorado River runs through, specifically those in the Mountain Time Zone.

2. **Identify Key Entities:** The key entities are "Colorado River" and "Mountain Time Zone".

3. **Find States the Colorado River Runs Through:**
- I look for triples involving "Colorado River" and a "partially_contained_by" or similar relationship with a state.
- `['Colorado River', 'location.location.partially_containedby', 'Arizona']` indicates the Colorado River runs through Arizona.
- `['Colorado River', 'location.location.partially_containedby', 'Colorado']` indicates the Colorado River runs through Colorado.
- `['Nevada', 'location.location.partially_contains', 'Colorado River']` indicates the Colorado River runs through Nevada.
- `['Utah', 'location.location.partially_contains', 'Colorado River']` indicates the Colorado River runs through Utah.

So, the states the Colorado River runs through are: Arizona, Colorado, Nevada, Utah.

4. **Find States in the Mountain Time Zone:**
- I look for triples involving "Mountain Time Zone" and a "time_zones" relationship with a state.
- `['Arizona', 'location.location.time_zones', 'Mountain Time Zone']` indicates Arizona is in the Mountain Time Zone.
- `['Colorado', 'location.location.time_zones', 'Mountain Time Zone']` indicates Colorado is in the Mountain Time Zone.
- `['Nevada', 'location.location.time_zones', 'Mountain Time Zone']` indicates Nevada is in the Mountain Time Zone.

So, the states in the Mountain Time Zone are: Arizona, Colorado, Nevada.

5. **Find the Intersection:** I need to find the states that appear in both lists.
- States the Colorado River runs through: Arizona, Colorado, Nevada, Utah.
- States in the Mountain Time Zone: Arizona, Colorado, Nevada.

The intersection of these two sets is: Arizona, Colorado, Nevada.

6. **Formulate the Answer:** The states that the Colorado River runs through and are in the Mountain Time Zone are Arizona, Colorado, Nevada.</think>
<answer>['arizona', 'colorado', 'nevada']</answer>

**Question**:What states does the Colorado River run through in the Mountain Time Zone?**Answer:["Colorado", "Nevada", "Arizona"]** **GCR**

"**Reasoning Paths**:\n#
Reasoning Path:\nMountain Time Zone -> common.topic.image -> Timezoneswest -> common.image.appears_in_topic_gallery
-> Samoa Time Zone\n# Answer:\nSamoa Time Zone\n#
Reasoning Path:\nColorado River -> geography.river.basin_countries -> United States of America ->
location.country.administrative_divisions -> Arizona\n# Answer:\nArizona\n#
Reasoning Path:\nColorado River -> geography.river.basin_countries -> United States of America ->
location.country.first_level_divisions -> Colorado\n# Answer:\nColorado\n#
Reasoning Path:\nColorado River -> location.location.partially_containedby -> Utah\n# Answer:\nUtah\n#
Reasoning Path:\nColorado River -> location.location.partially_containedby -> Colorado\n# Answer:\nColorado\n\n
The answer is **"Arizona  Colorado  Utah"**

Figure 14: Analysis of superlative reasoning patterns.

GED operates with less stringency than our path reward function; it allows for reward optimization merely through the addition or substitution of nodes and edges, rather than enforcing precise path alignment. In conclusion, adopting a more complex reward function with rich graph structural features does not necessarily guarantee superior model efficacy.

## M    PERFORMANCE ANALYSIS WITH LESS CAPABLE CoT MODELS

To investigate the impact of a weaker teacher model, we employed Qwen-32B as the teacher model in this section. As shown in the Table 11, although the weaker teacher resulted in a decline in SFT performance, the final EoG performance following RL training recovered to a level comparable to the original baseline. This demonstrates that our method is relatively insensitive to the teacher model's capabilities; rather, the performance gains are primarily driven by the RL training, as this stage effectively refines the policy through interaction with the ground-truth graph environment.

## N    STATISTICAL EVALUATION DETAILS

To ensure the robustness of our results, we conducted three independent runs of EoG using different random seeds. As shown in Table 12, by incorporating the standard deviation, we confirm that EoG maintains robust state-of-the-art performance (e.g., $81.5 \pm 0.33$ on average), unaffected by statistical fluctuations. This establishes it as a strong baseline and demonstrates the statistical reliability of our method.

Table 7: Computational Cost Comparison: SFT vs. RL. Due to difference among tokenizers used by different models, we measured SFT Computational Cost using SFT instances instead of tokens, as we consider instances to be a fairer indicator of the SFT budget.

| Metric | CWQ | WebQSP | GrailQA | 2WikiMultihopQA |
|---|---|---|---|---|
| ***Supervised Fine-Tuning (SFT)*** | | | | |
| Training Steps | 400 | 300 | 300 | 400 |
| Training Time (h) | 6.1 | 4.2 | 4.5 | 5.1 |
| Train Batch Size | 16 | 16 | 16 | 16 |
| Micro Batch Size | 2 | 2 | 2 | 2 |
| Total Instances | 3537 | 2344 | 4774 | 3102 |
| ***Reinforcement Learning (RL)*** | | | | |
| Training Steps | 108 | 36 | 12 | 96 |
| Training Time (h) | 509.6 | 119.2 | 76.8 | 205.6 |
| Batch Size | 256 | 256 | 256 | 256 |
| Mini Batch Size | 128 | 128 | 128 | 128 |
| Micro Batch Size | 2 | 2 | 2 | 2 |

Table 8: Comparison of the performance of different closed-source commercial models on the KGQA datasets under various temperature settings.

| Dataset | GPT-5 | | Gemini-2.5-flash | | Gemini-2.5-pro | |
|---|---|---|---|---|---|---|
| | T=0.2 | T=0.7 | T=0.2 | T=0.7 | T=0.2 | T=0.7 |
| WebQSP | 77.5 | 78.1 | 78.2 | 78.4 | 79.8 | 79.2 |
| CWQ | 67.6 | 67.6 | 59.3 | 59.7 | 65.3 | 65.7 |
| GrailQA | 85.4 | 85.2 | 83.8 | 84.3 | 84.5 | 85.1 |
| QALD10-en | 50.4 | 51.3 | 46.2 | 46.1 | 48.3 | 48.5 |
| 2WikiMultihopQA | 83.4 | 82.9 | 83.1 | 83.9 | 82.6 | 82.9 |

Beyond reporting mean performance scores, we further validated the reliability of our results through statistical hypothesis testing. Table 13 details the t-statistics and p-values comparing EoG with the GCR baseline. In all datasets, We performed a standard t-test, yielding a p-value $< 0.05$, which confirms that the performance gains of EoG over GCR are statistically significant and not due to random variance.

As illustrated in Figure 15, RL typically exhibits high variance during the training phase, as represented by the shaded error bands. It can be observed from Figure 15 that the model incorporating the path reward mechanism consistently achieves superior performance compared to the baseline across all three random seeds. The distinct separation between the error bands of the two methods after the 75th step indicates that our proposed framework achieves stable convergence and that the performance gain is statistically significant rather than a result of random variance.

## O  CONSTRUCTION OF GROUND-TRUTH REASONING PATHS

For datasets lacking explicit reasoning paths (e.g., WebQSP, CWQ, GrailQA, and QALD10-en), we constructed ground-truth paths ($r_g$) using a two-stage "Search-and-Verify" pipeline. This process bridges the gap in existing benchmarks by deriving reliable reasoning chains from raw Question-Answer (QA) pairs. **Path Retrieval via Breadth-First Search (BFS).** We first employed BFS on the Knowledge Graph to retrieve all potential topological paths connecting the topic entity (identified in the question) to the ground-truth answer entity. This step ensures high recall of potential reasoning chains. **Semantic Verification via LLM.** Since BFS often yields "spurious paths"—paths that are topologically valid but logically unrelated to the question—we utilized an LLM (Gemini-2.5-Flash)

Table 9: Detailed configuration of training and testing parameters for the GRPO phase.

| Category | Configuration |
|---|---|
| **Hardware** | Compute Hardware: $8 \times$ H100 GPUs |
| | GPU Memory Utilization: 0.6 |
| | FSDP Offload: Param & Optimizer |
| **Backbone** | Rollout Engine: vLLM |
| | Gradient Checkpointing: True |
| **Training** | Optimizer: Adam (implied by standard GRPO setup) |
| | Learning Rate: $4 \times 10^{-6}$ |
| | LR Warmup Steps: 10 |
| | Batch Size: 256 (Mini: 128, Micro: 3) |
| | Gradient Clip: 1.0 |
| | KL Loss Term: Disabled |
| | KL In Loss/Reward: False |
| | Epochs: 6 (Train) / 0 (Test) |
| **Sampling** | Temperature: 1.0 (Train) / 0.8 (Test) |
| | Top-p: 1.0 (Train) / 0.9 (Test) |
| | Sampling Number ($N$): 6 (Train) / 8 (Test) |
| **Constraints** | Max Prompt Length: 10,000 (Train) / 20,000 (Test) |
| | Max Response Length: 10,000 |
| | Overlong Penalty: $> 3,000$ tokens (Train) / Disabled (Test) |

Table 10: Performance on Various Datasets Across Different GED Coefficients

| Dataset | $\lambda = 0.2$ | | $\lambda = 0.4$ | | $\lambda = 0.6$ | | $\lambda = 0.8$ | | $\lambda = 1.0$ | |
|---|---|---|---|---|---|---|---|---|---|---|
| | F1 | Hit@1 | F1 | Hit@1 | F1 | Hit@1 | F1 | Hit@1 | F1 | Hit@1 |
| CWQ | 69.7 | 78.2 | 68.5 | 77.3 | 67.9 | 77.1 | 67.1 | 76.3 | 67.3 | 75.9 |
| WebQSP | 80.6 | 89.6 | 79.4 | 88.9 | 80.2 | 89.3 | 79.8 | 89.2 | 78.9 | 88.2 |
| GrailQA | 90.3 | 91.8 | 89.9 | 91.4 | 89.6 | 91.2 | 90.1 | 91.3 | 89.7 | 91.5 |
| 2WikiMultihopQA | 84.2 | 84.8 | 83.7 | 84.2 | 83.5 | 83.9 | 83.3 | 83.7 | 83.1 | 83.5 |

Table 11: Performance of EoG on the CWQ and WebQSP Datasets, using Qwen3-32B as the teacher model.

| Dataset | Metric | EoG (SFT Only) | EoG (SFT + RL) | Improvement (%) |
|---|---|---|---|---|
| CWQ | F1 Score | 55.8 | 70.7 | -4.3 |
| | Hit@1 | 61.5 | 75.6 | -8.4 |
| WebQSP | F1 Score | 71.3 | 80.2 | -5.4 |
| | Hit@1 | 82.6 | 87.9 | -4.9 |

Table 12: Performance comparison between our EoG framework (Llama-3.1-8B) and state-of-the-art closed-source models. The bottom section highlights the significant improvement of our RL-finetuned model over its SFT model. We report the mean ± standard deviation for our method.

| Method | Model | WebQSP | | CWQ | | GrailQA | | QALD10-en | | 2WikiMultihopQA | |
|---|---|---|---|---|---|---|---|---|---|---|---|
| | | Hit@1 | F1 | Hit@1 | F1 | Hit@1 | F1 | Hit@1 | F1 | Hit@1 | F1 |
| [†]Gemini-2.5 Flash | - | 91.8 | 78.2 | 65.5 | 59.3 | 90.3 | 83.8 | 56.7 | 46.2 | 83.9 | 83.1 |
| [†]Gemini-2.5 Pro | - | 92.1 | 79.8 | 71.9 | 65.3 | 91.6 | 84.5 | 58.6 | 48.3 | 85.1 | 82.6 |
| [†]GPT-5 | - | 86.1 | 77.5 | 74.1 | 67.6 | 90.5 | 85.4 | 59.2 | 50.4 | 84.2 | 83.4 |
| $EoG_{SFT}$ | Qwen2.5-7B | 83.9 | 72.6 | 68.3 | 60.3 | 89.2 | 87.6 | 55.6 | 44.1 | 82.5 | 81.9 |
| | Llama-3.1-8B | 86.3 | 74.5 | 70.5 | 62.1 | 91.4 | 88.2 | 57.1 | 48.7 | 83.1 | 82.7 |
| EoG | Qwen2.5-7B | 90.7 ± 0.34 | 78.1 ± 0.45 | 78.7 ± 0.26 | 69.8 ± 0.37 | 91.7 ± 0.29 | 88.5 ± 0.43 | 67.3 ± 0.17 | 57.8 ± 0.32 | 83.9 ± 0.47 | 82.9 ± 0.24 |
| | Llama-3.1-8B | 92.8 ± 0.28 | 81.3 ± 0.33 | 82.6 ± 0.41 | 73.9 ± 0.20 | 92.1 ± 0.30 | 90.6 ± 0.25 | 70.6 ± 0.49 | 61.9 ± 0.38 | 85.3 ± 0.23 | 84.3 ± 0.44 |

Table 13: Statistical comparison between GCR and our EoG framework using a one-sample t-test ($df = 4$). The table reports the baseline score, our model's performance (Mean ± SD), and the calculated $t$-statistics and $p$-values. Results indicate statistically significant improvements across all metrics.

| Dataset | Metric | GCR (Baseline) | EoG (Ours) (Mean ± SD) | $t$-value | $p$-value |
|---|---|---|---|---|---|
| WebQSP | Hit@1 | 92.2 | 92.8 ± 0.28 | 4.79 | 0.009 |
| | F1 | 79.1 | 81.3 ± 0.33 | 14.91 | < 0.001 |
| CWQ | Hit@1 | 75.8 | 82.6 ± 0.41 | 37.08 | < 0.001 |
| | F1 | 61.7 | 73.9 ± 0.20 | 136.46 | < 0.001 |
| GrailQA | Hit@1 | 88.4 | 92.1 ± 0.30 | 27.57 | < 0.001 |
| | F1 | 85.1 | 90.6 ± 0.25 | 49.20 | < 0.001 |
| QALD10 | Hit@1 | 57.6 | 70.6 ± 0.49 | 59.33 | < 0.001 |
| | F1 | 49.3 | 61.9 ± 0.38 | 74.12 | < 0.001 |
| 2WikiMultihopQA | Hit@1 | 84.6 | 85.3 ± 0.23 | 6.80 | 0.002 |
| | F1 | 77.5 | 84.3 ± 0.44 | 34.56 | < 0.001 |

as a semantic filter. The LLM was prompted to evaluate whether each retrieved path logically corresponds to the semantic intent of the natural language question. Only paths that passed this semantic verification were retained as $r_g$ for calculating $R_{path}$.

Consequently, this approach provides the necessary supervision signals for the second-stage RL training, addressing the lack of $r_g$ in open-source benchmarks.

## P    DETAILED DEFINITION OF EXPLORATION METRICS

To provide a fine-grained analysis of exploration behavior, we parse the triples mentioned in the model's reasoning trace (the `<think>` block) on the CWQ test set.

Let $\mathcal{D}$ denote the test set with size $N = |\mathcal{D}|$. For the $i$-th question, let $T_{pred}^{(i)}$ be the set of valid unique triples extracted from the model's reasoning path, and $T_{gold}^{(i)}$ be the set of triples in the ground-truth reasoning path.

**Exploration Efficiency** (↓). This metric measures the search overhead, representing the average number of triples the model explores to identify one correct reasoning triple. A lower value indicates higher efficiency, implying less wasted exploration effort for each valid discovery. It is calculated as the mean ratio of total extracted triples to the number of correct triples found:

$$\text{Exploration Efficiency} = \frac{1}{N} \sum_{i=1}^{N} \frac{|T_{pred}^{(i)}|}{|T_{pred}^{(i)} \cap T_{gold}^{(i)}|} \tag{6}$$

**Coverage of the Reasoning Space** (↑). To define this metric, we first conceptualize the Reasoning Space for a given question as the set of all potential valid paths within the Knowledge Graph that

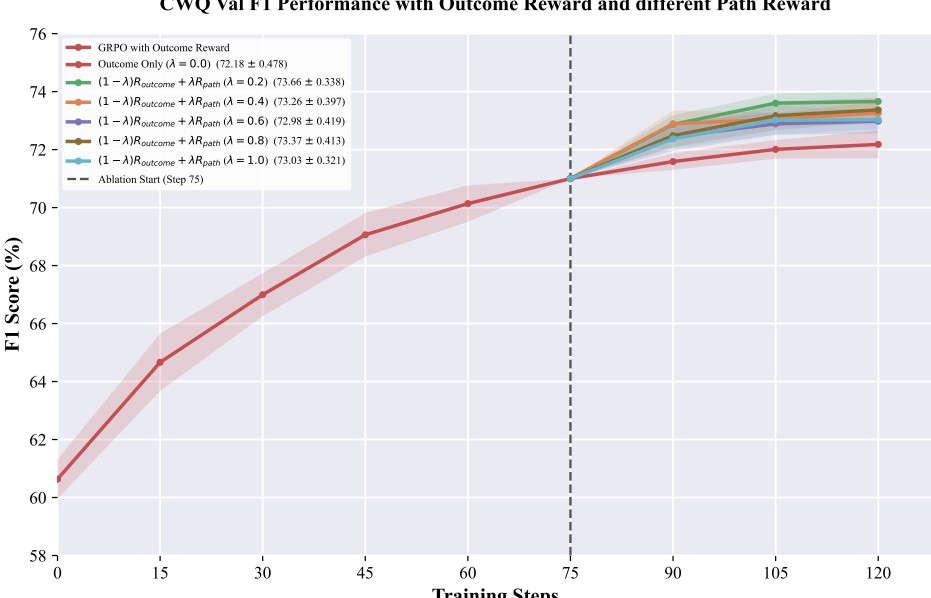

Figure 15: Validation F1 performance on the CWQ dataset with varying Path Reward weights. The total reward is formulated as $R = (1 - \lambda)R_{outcome} + \lambda R_{path}$, where $\lambda$ denotes the coefficient shown in the legend. The dashed vertical line marks the start of the second training phase at step 75, where the reward shifts from result-only RL to a mixed result-and-path reward. Solid lines and shaded regions represent the mean and standard deviation across 3 random seeds, respectively. The results indicate that $\lambda = 0.2$ achieves the optimal balance, yielding the highest F1 score and robust stability compared to the baseline ($\lambda = 0.0$).

connect the starting entity to the target answer entities. Consequently, this metric measures the reasoning recall, quantifying the average proportion of the ground-truth logical chain (a subset of the reasoning space) that is successfully uncovered by the model's exploration. A higher value indicates better coverage, meaning the model finds more of the required logical steps. It is calculated as:

$$\textbf{Coverage} = \frac{1}{N} \sum_{i=1}^{N} \frac{|T_{pred}^{(i)} \cap T_{gold}^{(i)}|}{|T_{gold}^{(i)}|} \tag{7}$$

