# OpenReview forum: "Explore-on-Graph: Incentivizing Autonomous Exploration of Large Language Models on Knowledge Graphs with Path-refined Reward Modeling"
_ICLR.cc/2026/Conference — ICLR 2026 Poster_

### Official Review · Reviewer_Fb3S · 2025-10-28

**Soundness:** 3
**Presentation:** 2
**Contribution:** 2
**Rating:** 4
**Confidence:** 3

**Summary:**

This paper addresses the challenge of poor generalization in existing Large Language Model (LLM) methods for Knowledge Graph Question Answering (KGQA), which often fail when faced with reasoning patterns not seen during training. The authors propose "Explore-on-Graph" (EoG), a novel framework designed to incentivize LLMs to autonomously explore diverse reasoning paths on a knowledge graph (KG).

**Strengths:**

The core contribution—incentivizing autonomous exploration on KGs via reinforcement learning—is a novel and important step beyond prevalent imitation-based methods. While RL has been used for KG reasoning in the past, its application to modern LLMs with the proposed two-phase reward structure (outcome + path-refinement) is original. The "path-refined reward" is an intuitive and clever mechanism to make the exploration process more efficient and meaningful, rather than just rewarding the final outcome. This directly addresses the critical problem of out-of-distribution generalization in KG reasoning.

**Weaknesses:**

The path-refined reward, Rpath, is a key component of the method. Its calculation (Equation 4) requires a "ground-truth reasoning path" rg. However, the paper fails to explain how these ground-truth paths are obtained for the training data across all five datasets. While some datasets (e.g., 2WikiMultihop) may provide such paths, it is not standard for others like CWQ, WebQSP, or GrailQA. This is a critical missing methodological detail.

The proposed SFT + RL pipeline, particularly using GRPO with multiple rollouts per prompt, appears to be computationally very expensive. The paper mentions using 8xH100 GPUs but does not provide a broader discussion on the training time, total computational budget, or the trade-offs between performance gains and the significant training overhead. A comparison of the computational cost against simpler fine-tuning methods would be valuable for assessing the practical applicability of EoG.

**Questions:**

see weaknesses

---

> ### Author Response · Authors · 2025-11-21
> **Feedback to Official Review of Reviewer Fb3S**
>
> ### **Response to Weakness 1**
>
> For datasets that do not explicitly provide reasoning paths (WebQSP, CWQ, GrailQA, QALD10-en),
> we constructed the ground-truth paths $r_g$ using a "Search-and-Verify" pipeline:
> - Path Retrieval via Breadth-First Search (BFS): We first employed BFS on the Knowledge Graph to retrieve all potential paths connecting the topic entities(identified in the question) to the ground-truth answer entities.
> - Semantic Verification via LLM: Since BFS may yield "spurious paths" (paths that are topologically connected but semantically unrelated to the question logic), we utilized an LLM (Gemini-2.5-Flash) to evaluate the retrieved paths.
> The LLM was prompted to verify whether each path logically corresponds to the semantic intent of the natural language question.
> Only paths that passed this semantic verification were retained as the ground-truth paths $r_g$ for calculating $R_{path}$.
>
> Overall, this approach effectively bridges the gap in existing benchmarks by deriving reliable reasoning chains, enabling us to train faithful reasoning policies even in the absence of human-annotated paths.  We have updated Section 3.2.2 in the submitted revision to explicitly describe the pipeline for constructing ground-truth paths.
>
> ### **Response to Weakness 2**
> We sincerely thank the reviewer for raising this crucial question regarding the computational cost of the SFT + RL pipeline.
> We have added a detailed cost breakdown in Appendix I. Specifically, the average training time across all datasets was 227.8 GPU hours (approximately $2519.5 on [Google Cloud](https://cloud.google.com/products/calculator) ). We believe this overhead is highly justified for two key reasons:
> - Simpler fine-tuning hits a hard performance ceiling. As shown in Table 1, SFT achieves only 62.1% F1 on CWQ, whereas our RL approach boosts this to 73.9%. The computational investment is necessary to achieve this +11.8% absolute improvement that simpler methods cannot deliver.
>
> - Unlike agent-based baselines that require multiple expensive LLM calls per question during inference, our trained model generates reasoning paths in a single pass, making it highly efficient for practical deployment.
>
>
> We are particularly grateful for your insightful questions. These inquiries prompted us to conduct a deeper analysis, leading to a more transparent and comprehensive evaluation of our method. We remain committed to continuous improvement and are open to any further discussion to clarify remaining details. We look forward to engaging further with you.

---

### Official Review · Reviewer_dVt9 · 2025-10-28

**Soundness:** 2
**Presentation:** 3
**Contribution:** 2
**Rating:** 4
**Confidence:** 4

**Summary:**

This paper proposes Explore-on-Graph (EoG) framework that incentivizes LLMs to autonomously explore diverse reasoning paths on Knowledge Graphs for Question Answering. The methods consist two stages: (i) SFT using long CoT by Gemini 2.5 Flash, (ii) reinforcement learning with Group Relative Policy Optimization using both outcome reward and path refined reward. The authors evaluate EoG on five KGQA benchmarks and demonstrate state-of-the-art performance, though I must say the results for the baseline are mostly missing.

**Strengths:**

1. This method outperforms powerful closed-source models such as GPT-5 and Gemini-2.5 Pro, which is impressive.
2. The paper is well written, with clear motivation and problem formulation, and provides a good example (Figure 1) illustrating the limitation of existing approaches and the EoG methods.
3. Smaller open-source LLM trained by EoG can compete with larger closed-source ones, which means EoG addresses some of the current compute resource limitations.
4. Well-organized experiment, ablation study effectively demonstrates the importance of each component.

**Weaknesses:**

1. Limited technical novelty: The core contribution combines existing techniques (SFT, GRPO, simple reward design) without significant algorithmic innovation. The path reward is particularly simplistic, using only substring matching. In the area of KG reasoning, using reinforcement learning to explore path is a general and common practice. Check MINERVA [1] and DeepPath [2].

2. The reliance on Gemini 2.5 Flash for dataset generation creates a dependency that may limit reproducibility. The paper doesn't discuss alternatives or provide the generated datasets.

3. More than half of the baseline experimental results in Table 1 are missing, and the authors do not provide any explanation in the paper. I do not think that the effectiveness of EoG can be proven based on Table 1 alone, as the baseline results are extremely limited.

4. The paper lacks a discussion of training costs, convergence time, and computational requirements for the RL stage.  As the paper says it trains EoG on 8*H100 GPUs, which is very costly. This may be a strong limitation of the proposed method.

5. The substring-based matching in Equation 4 is brittle and may not capture semantic equivalence, paraphrases, or partial correctness in reasoning paths.


[1] Das, Rajarshi, et al. "Go for a Walk and Arrive at the Answer: Reasoning Over Paths in Knowledge Bases using Reinforcement Learning." International Conference on Learning Representations. 2018.
[2] Xiong, Wenhan, Thien Hoang, and William Yang Wang. "DeepPath: A Reinforcement Learning Method for Knowledge Graph Reasoning." Proceedings of the 2017 Conference on Empirical Methods in Natural Language Processing. 2017.

**Questions:**

1. What are the computational costs of the RL training stage compared to standard supervised fine-tuning? How many GPU hours are required for convergence?
2. Have you considered more sophisticated path reward designs that account for semantic similarity rather than exact substring matching? For example, using graph edit distance or learned similarity metrics?
3. Can you provide the generated CoT datasets or detailed statistics about them to enable reproducibility without access to Gemini 2.5 Flash?
4. How sensitive is the method to the quality of the initial SFT stage? What happens if a weaker teacher model is used for data generation?
5. Have you analyzed failure cases where exploration leads to incorrect paths? How does the model handle ambiguous questions where multiple valid reasoning paths exist?
6. Could you provide theoretical analysis or empirical evidence about the exploration efficiency and coverage of the reasoning space? For example, to improve the content of case study?
7. Why are so many experimental results missing in Table 1? (I don’t think “the original paper did not report the experimental results of this dataset” is a good answer)

---

> ### Author Response · Authors · 2025-11-21
> **Feedback to Official Review of Reviewer dVt9(Part I)**
>
> ### **Response to Weakness 1**
> We fully agree that using RL for path exploration in KGs is a well-established field.
> However, we respectfully clarify a fundamental distinction in the problem definition, which defines our technical novelty.
> - MINERVA/DeepPath operates in a discrete action space, where the agent explicitly selects the next node/edge.
> They focus on path-finding within the graph structure.
>  EoG operates in a generative action space. Our agent generates continuous Chain-of-Thought (CoT) tokens.
>  We use RL not just to traverse the graph, but to optimize the semantic reasoning policy of the LLM.
>  This allows EoG to solve complex queries that require linguistic understanding, which traditional graph-walking agents cannot handle.
> - Response regarding Path Reward Design: We choose this design because faithfulness is a critical aspect of KGQA tasks. While path matching algorithms based on semantic similarity or Graph Edit Distance can better capture linguistic variations, they often lead to reasoning processes that cannot be executed on the KG (e.g., schema mismatches), resulting in hallucinations and degraded performance. Furthermore, since GRPO training requires continuous collection of reasoning experiences, complex algorithms like Graph Edit Distance often face significant efficiency challenges. Therefore, we opted for this simple but effective path reward to ensure both executability and training feasibility.
>
> - While SFT+RL is a known pattern, its application to make an LLM autonomously explore a KG (beyond just imitating SFT paths) is non-trivial. Our core contribution is the framework that enables this exploration, moving beyond static imitation learning which, as we show, fails on O.O.D. patterns.
>
>
> ### **Response to Weakness 2 & Question 3**
> To ensure full reproducibility, we have submitted the complete datasets generated by Gemini 2.5 Flash in the supplementary materials. For detailed statistics and the construction process of the SFT dataset, please refer to Appendix J.
> ### **Response to Weakness 3 & Question 7**
> We reproduce and report results for the strongest state-of-the-art methods GCR and DoG on Wikidata-based datasets to fill existing gaps.
> These updated comparisons are detailed in the revised Table 1.
> Crucially, EoG consistently outperforms these strong baselines and confirms our state-of-the-art performance.
>
> The remaining omissions are due to fundamental schema incompatibilities between Freebase-centric baselines and Wikidata benchmarks (QALD10-en, 2WikiMultihop).
> For instance, methods like RoG rely on checkpoints aligned with Freebase relation vocabularies.
> As explicitly noted by DoG[1], RoG "cannot be applied on 2Wikimultihop, as its checkpoint trained on Freebase fails to generate relation labels of Wikidata".
> Reproducing such methods would require re-engineering their retrieval modules and re-training from scratch, introducing significant implementation bias.
> We excluded these invalid combinations to ensure the fairness and validity of the comparison.
>
> ### **Response to Weakness 4 & Question 1**
> We have explicitly detailed the training costs, convergence time, and computational requirements for the RL stage in Appendix I. Specifically, the total training time on the CWQ dataset amounted to 509.6 GPU hours.
> Based on current [Google Cloud](https://cloud.google.com/products/calculator) pricing for 8 $\times$ H100 instances, this corresponds to a total cost of approximately 5,638.13 dollars, which we consider reasonable given the substantial performance improvements.
>  The use of 8 $\times$ H100 GPUs was chosen primarily to accelerate our experimental iteration, not as a strict hardware requirement. Techniques such as Gradient Checkpointing[2] and ZeRO-Offload (e.g., DeepSpeed[3]/FSDP[4])  allow for efficient memory management, ensuring that our method can be adapted to run on devices with lower VRAM (e.g., A100s ).

---

> ### Author Response · Authors · 2025-11-21
> **Feedback to Official Review of Reviewer dVt9(Part II)**
>
> ### **Response to Weakness 5**
> For clarity, we further add more details to explain the design of the path reward.
> Equation 4 does not enforce a rigid, sentence-level string comparison.
> Instead, it functions as a structure-aware verification mechanism.
> Specifically, we calculate the reward by extracting ground-truth triples in the form of [subject, relation, object], and a triple is considered correct only when all three atomic elements appear simultaneously within the generated reasoning text.
> - Semantic Equivalence: Our model does not penalize the use of synonyms in the  `<think>` process. The model is encouraged to leverage its semantic understanding (e.g., inferring that "birthplace" maps to people.born_in) to bridge logical steps. However, to receive the reward, this semantic understanding must eventually be grounded into the exact schema identifiers. This design balances flexible semantic reasoning with strict graph executability.
> - Paraphrases: The reward allows our model to weave components into various linguistic structures (e.g., active/passive voice), mitigating brittleness regarding sentence-level paraphrasing.
> - Partial Correctness: The reward is calculated as the ratio of matched triples. This naturally captures and rewards partial correctness in reasoning paths (e.g., correctly identifying 2 out of 3 steps) rather than requiring an all-or-nothing match.
>
> ### **Response to Question 2**
> We implement the Graph Edit Distance (GED) score to reward the reasoning paths. As indicated in the
> Table 10 in Appendix L, the utilization of GED failed to enhance the model’s performance on the test set. For instance, on the CWQ dataset, the F1 score peaks at only 69.7 with $\lambda=0.2$ and further declines to 67.3 as $\lambda$ increases to 1.0. A similar downward trend is observed on WebQSP, where the F1 score drops from 80.6 ($\lambda=0.2$) to 78.9 ($\lambda=1.0$).
> We attribute this phenomenon to two primary factors.
>
> First, since we explicitly constrained the reasoning structure during Supervised Fine-Tuning (SFT), our
> original path-based reward function proved to be simple but effective within this context.
> Second,
> GED operates with less stringency than our path reward function; it allows for reward optimization
> merely through the addition or substitution of nodes and edges, rather than enforcing precise path
> alignment.
>
> Besides, our framework employs Group Relative Policy Optimization (GRPO), which requires sampling multiple reasoning paths (e.g., 8 samples) per query to estimate advantages. Calculating GED is computationally intensive compared to the $O(N)$ complexity of substring matching.
>
> In conclusion, adopting a more complex reward function with rich graph structural
> features does not necessarily guarantee superior model efficacy.
>
>
>
> ### **Response to Question 4**
> we conducted a new experiment using a weaker teacher model Qwen3-32B[5] to answer the question regarding the sensitivity of our method to the quality of the SFT stage.
> We constructed  new WebQSP and CWQ SFT datasets of the exact same size as the original version and fully reproduced the training pipeline.
> The results are shown below:
> | Dataset | Teacher Model                | SFT Performance (F1) | Final EoG Performance (F1) |
> |---------|-----------------------------|-------------------------------|------------------------------------------|
> | WebQSP  | Gemini 2.5 Flash  | 74.5              | 81.3                            |
> |         | Qwen3-32B         | 71.3                | 80.2                  |
> | CWQ     | Gemini 2.5 Flash  | 62.1                    | 73.9                                 |
> |         | Qwen3-32B          | 55.8                | 70.7                  |
>
> As shown in the table, although the weaker teacher (Qwen3-32B) resulted in a drop in SFT performance, the final EoG performance after RL training recovered to a level comparable to the original baseline. This confirms that our method is insensitive to the teacher's reasoning quality, as the RL stage effectively refines the policy through interaction with the ground-truth graph environment.

---

> ### Author Response · Authors · 2025-11-21
> **Feedback to Official Review of Reviewer dVt9(Part III)**
>
> ### **Response to Question 5**
> Exploration Failure Analysis: We categorized the generated responses into Success Samples (F1=1) and Failure Samples (F1<1). For each category, we calculated the Hallucination Rate, defined as the percentage of generated reasoning triples $(s, r, o)$ in the `<think>` texts that do not exist in the knowledge graph.
>
> | Model |  Hallucinated Rate(Overall) | Hallucinated Rate(Success sets)  | Hallucinated Rate(Failure sets) |
> |------|-------|-------|-------|
> $\text{EoG}_{\text{SFT}}$ | 16.11 |14.02 | 18.02
> $\text{EoG}_{\text{Outcome}}$ | 11.63 |9.43 |14.16
> $\text{EoG}$ | 11.75 |9.99 |13.84
>
>
> - Contrary to the concern that autonomous exploration might lead to uncontrolled hallucinations, our RL-based approach significantly reduces the hallucination rate in failure cases compared to the SFT baseline (dropping from 18.02% to 13.84%). This indicates that RL incentivizes the model to adhere more strictly to the valid topology of the Knowledge Graph.
> -  The lower hallucination rate in $\text{EoG}$'s incorrect samples (13.84%) compared to $\text{EoG}_{\text{Outcome}}$ (14.16%) suggests that the Path-Refined Reward effectively acts as a soft constraint. Even when our model fails to find the correct answer, its reasoning paths remain structurally valid and grounded, ensuring that failures are primarily due to logical selection errors rather than factual hallucinations.
>
> Ambiguous Questions Handling: The model effectively manages ambiguous questions with multiple valid reasoning paths by utilizing an F1-based outcome reward signal that incentivizes high recall to capture the complete set of correct answers.
> This is reinforced by prompt engineering that explicitly instructs the model to systematically analyze multiple starting entities and consider all possible solutions.
> Besides, GRPO that optimizes the policy to navigate a broader exploration space, ensures the activation of diverse and valid reasoning pathways.
>
> ### **Response to Question 6**
> To evaluate the exploration efficiency and coverage of the reasoning space, we parsed the triples mentioned in the think text of our models on the CWQ test set.
> Let $\mathcal{D}$ denote the test set with size $N = |\mathcal{D}|$.
> For the $i$-th question,
> let $T_{pred}^{(i)}$ be the set of valid unique triples extracted from the model's reasoning path, and $T_{gold}^{(i)}$ be the set of triples in the ground-truth (golden) reasoning path.
> We quantified these metrics as the mean values over the test set, defined as follows:
>
> Exploration Efficiency: This metric measures the search overhead, representing the average number of triples the model explores to identify one correct reasoning triple. It is calculated as the mean ratio of total extracted triples to the number of correct triples found:
> $$\text{Exploration Efficiency} = \frac{1}{N} \sum_{i=1}^{N} \frac{|T_{pred}^{(i)}|}{|T_{pred}^{(i)} \cap T_{gold}^{(i)}|}
> $$
> Note: A lower value indicates higher efficiency, implying less wasted exploration effort for each valid discovery.
>
> Coverage of the Reasoning Space: To define this metric, we first conceptualize the Reasoning Space for a given question as the set of all potential valid paths within the Knowledge Graph that connect the starting entity to the target answer entities. Consequently, this metric measures the reasoning recall, quantifying the average proportion of the ground-truth logical chain (a subset of the reasoning space) that is successfully uncovered by the model's exploration. It is calculated as:
> $$\text{Coverage} = \frac{1}{N} \sum_{i=1}^{N} \frac{|T_{pred}^{(i)} \cap T_{gold}^{(i)}|}{|T_{gold}^{(i)}|}$$
> Note: A higher value indicates better coverage, meaning the model finds more of the required logical steps.
>
> | Model | Exploration Efficiency | Coverage of the Reasoning Space  |
> |------|-------|-------|
> $\text{EoG}_{\text{SFT}}$ | 2.877 |0.615
> $\text{EoG}_{\text{Outcome}}$ |3.028  |0.689
> $\text{EoG}$ | 2.887 |0.723
>
> The results clearly demonstrate the impact of our Path-Refined Reward:
> - High Coverage: $\text{EoG}$ achieves the highest coverage (0.723), significantly outperforming $\text{EoG}_{\text{SFT}}$ (0.615), which confirms that RL incentivizes the model to explore deeper reasoning paths that SFT misses.
> - Maintained Efficiency: While pure outcome-based RL ($\text{EoG}_{\text{Outcome}}$) increases coverage, it degrades efficiency (rising to 3.028), indicating noisy exploration. In contrast, $\text{EoG}$ maintains an efficiency score (2.887) comparable to the SFT baseline (2.877) while maximizing coverage. This proves that our path reward effectively prunes futile search branches, ensuring exploration is both broad and efficient.
>
> We sincerely thank the reviewer for your rigorous and insightful feedback, which have significantly strengthened the technical depth and empirical robustness of our work.
> We hope these changes can address your concerns and highlight the unique value of EoG in the field of KGQA.

---

> ### Author Response · Authors · 2025-11-21
> **Feedback to Official Review of Reviewer dVt9(Part IV)**
>
> Refs:
>
> [1] Li, Kun, et al. "Decoding on graphs: Faithful and sound reasoning on knowledge graphs through generation of well-formed chains." Proceedings of the 63rd Annual Meeting of the Association for Computational Linguistics (Volume 1: Long Papers). 2025.
>
> [2] Chen, Tianqi, et al. "Training deep nets with sublinear memory cost." arXiv preprint arXiv:1604.06174 (2016).
>
> [3] Rasley, Jeff, et al. "Deepspeed: System optimizations enable training deep learning models with over 100 billion parameters." Proceedings of the 26th ACM SIGKDD international conference on knowledge discovery & data mining. 2020.
>
> [4] Zhao, Yanli, et al. "PyTorch FSDP: Experiences on Scaling Fully Sharded Data Parallel." Proceedings of the VLDB Endowment 16.12 (2023): 3848-3860.
>
> [5] Yang, An, et al. "Qwen3 technical report." arXiv preprint arXiv:2505.09388 (2025).

---

> ### Comment · Reviewer_dVt9 · 2025-11-26
>
> I appreciate the authors' efforts in responding to the weaknesses and replying to the questions. I have increased my score. Still have one concern that the GPU cost is too high to optimize the models with RL.

---

> > ### Author Response · Authors · 2025-11-26
> >
> > We sincerely thank the reviewer for the timely follow-up and for increasing the evaluation score. We fully understand the concern regarding the GPU cost of RL optimization. In our future work, we will focus on the following two directions to reduce the computational cost:
> > - Based on our O.O.D. experiments, we plan to leverage the RL-optimized model to collect high-quality reasoning trajectories. These high-quality reasoning samples can then be incorporated into the SFT stage of subsequent training rounds, enabling the model to encounter a more diverse set of reasoning patterns during SFT. This improves the model's exploration capability before the next round's RL stage begins and reduces the convergence time and GPU cost the new training round. By iterating this process, the SFT data for each new training round incorporates the exploratory capabilities activated by the previous round of RL training. This enables the model to improve iteratively, and correspondingly, the convergence time and GPU cost of RL in new optimization rounds decrease over iterations. As a result, the overall cost of RL training remains within a reasonable range, avoiding the need to pay such a high cost every time the model is updated.
> > - We will integrate RL algorithms with higher efficiency to improve convergence speed and reduce resource consumption during training.
> >
> > Once again, we appreciate your constructive feedback and your recognition of our work.

---

### Official Review · Reviewer_Gg3c · 2025-10-29

**Soundness:** 3
**Presentation:** 3
**Contribution:** 3
**Rating:** 8
**Confidence:** 4

**Summary:**

The paper proposes Explore on Graph (EoG), where not only the correct answer but also the reasoning path is rewarded via reinforcement learning, enabling the model to explore novel reasoning paths that fall outside the distribution of pre-defined rules or supervised fine-tuning data.

The experiments are comprehensive and show strong results on five KGQA datasets, outperforming not only open-source but also even closed-source LLMs.

**Strengths:**

1. The motivation is well grounded. Not only the answer but also the reasoning path can server as good reward signals.
2. The experiments show strong results. For example, table 1 show EoG outperforms not only open-source but also even closed-source LLMs, and table 4 show strong results on OOD settings. The improvement is significant.

**Weaknesses:**

1. Some implementation details are not clear. Are phase 1: Outcome Reward Modeling and phase 2: Path-refined Reward Modeling implemented sequentially or simultaneously (as in Equation 5)?
2. Reproducibility: The code is currently unavailable, which hinders verification, reproduction, and improvement efforts. Open-source code is crucial for these processes.

**Questions:**

1. Same as Weakness 1: Are phase 1: Outcome Reward Modeling and phase 2: Path-refined Reward Modeling implemented sequentially or simultaneously (as in Equation 5)?
2. For the OOD experiment (Figure 5), what do the x and y axes represent? Is the model trained on the dataset on the x-axis and then evaluated on the dataset on the y-axis, or vice versa?
3. For the OOD experiment (Figure 5), what is the performance of other models in the OOD settings? It would be useful to compare EoG not only with EoG-SFT but also with other SOTA models (as in Table 1).
4. Some things can be improved for clarity and readability. For example, in Figure 3, the meanings of the x and y axes are only available in the main body text. The reading experience would be greatly enhanced if the x and y axes were directly labeled in the figure.

---

> ### Author Response · Authors · 2025-11-21
> **Feedback to Official Review of Reviewer Gg3c**
>
> ### **Response to Weakness 1 & Question 1**
> We apologize for the lack of clarity regarding the implementation details.
> Phase 1 (Outcome Reward Modeling) and Phase 2 (Path-refined Reward Modeling) are implemented sequentially.
> The training process proceeds as follows: We first train the model using only the outcome reward ($R_{outcome}$), as described in Section 3.2.1.
> This phase focuses on incentivizing the model to discover the correct answers, establishing a baseline capability for logical deduction without constraining the path too early.
> After Phase 1 converges, we initiate Phase 2 using the joint reward defined in Equation 5 ($R_{joint} = R_{outcome} + \alpha R_{path}$).
> In this second phase, we retain the outcome reward to maintain answer accuracy while introducing the path reward ($R_{path}$) to guide the model toward more efficient and semantic reasoning paths.
>
> To clarify this concept, we have updated the description in Section 3.2 in the revised paper to explicitly state the sequential training process.
>
>
> ### **Response to Weakness 2**
> We completely agree with the reviewer that open-source code is essential for verification and future research.
> To address this, we have uploaded the complete anonymous source code in the supplementary material for this rebuttal. The provided codebase includes:
> - Data Processing: Scripts for constructing the Long CoT datasets and processing the five benchmark datasets.
> - Training Scripts: The implementation of the Supervised Fine-Tuning (SFT) stage and the Reinforcement Learning stage (GRPO with outcome and path-refined rewards).
> - Evaluation: The evaluation scripts used to reproduce the main results and the ablation studies.
>
> Furthermore, we are committed to releasing the code and processed datasets via a public GitHub repository to support the community.
>
> ### **Response to Question 2**
> Thank you very much for highlighting the lack of clarity regarding the axes in Figure 5. We sincerely apologize for this oversight and any resulting confusion.
> Here is the clarification for the axes in Figure 5 (OOD experiment):
> - The Y-axis shows the dataset the model was trained on.
> - The X-axis shows the dataset the model was evaluated on.
>  We have fixed this issue in the revised paper by making the caption of Figure 5 much clearer and more explicit. We appreciate you identifying this important detail.
>
> ### **Response to Question 3**
> We have conducted additional experiments comparing our method with GCR and include the detailed results in Appendix N of the revised paper. The results demonstrate that EoG consistently outperforms GCR across all metrics. Specifically, while GCR relies on graph constraints to ensure path validity, it lacks the active exploration capability to discover novel reasoning patterns for out-of-distribution questions. In contrast, our RL-driven approach incentivizes the model to explore and optimize these paths, leading to superior performance.
>
> ### **Response to Question 4**
> We are highly grateful for your feedback regarding the clarity and readability of our figures.
> In the revised version, Figure 3 has clear, direct labels for both axes, eliminating the need to cross-reference with the main text.
>
> We sincerely thank the reviewer for the constructive and detailed feedback, which has significantly improved the clarity, reproducibility, and rigor of our work.

---

### Official Review · Reviewer_Fvcs · 2025-11-03

**Soundness:** 3
**Presentation:** 3
**Contribution:** 2
**Rating:** 6
**Confidence:** 3

**Summary:**

The paper proposes **Explore-on-Graph (EoG)**, a KG-augmented LLM reasoning framework that couples (i) supervised fine-tuning on long chain-of-thought traces and (ii) reinforcement learning with a **path-refined reward**. The core idea is to *incentivize autonomous exploration* of novel multi-hop paths on knowledge graphs, improving generalization beyond rule/imitation patterns. Concretely, the RL stage optimizes a GRPO-style objective using a final-answer **outcome reward** (entity-level F1 extracted from the `<answer>` tag) and an auxiliary **path reward** that measures how many ground-truth triples appear in the model’s `<think>` text; a weighted sum defines the joint reward. Experiments on **WebQSP, CWQ, GrailQA, QALD-10-en, and 2WikiMultihop** show consistent gains over recent KG-enhanced systems, and ablations indicate the path-refined reward materially contributes beyond SFT alone.

**Strengths:**

* **Clear motivation & problem framing.** The paper articulates why rule/imitation approaches struggle on OOD patterns and positions exploration as the missing capability. Figure 1 illustrates this vividly.
* **Method is simple, modular, and reproducible in principle.** The rewards (answer F1; path triple-match ratio) are transparent and plug into a standard GRPO objective with KL control.
* **Strong empirical results across diverse KGQA datasets** with consistent gains vs. strong baselines; ablations show each component’s contribution and explore the α trade-off between outcome and path rewards.
* **Ablations that are decision-useful.** Removing SFT, outcome, or path rewards degrades performance in expected ways, supporting the design choices. (Table 2 & ratio analyses in the text.)

**Weaknesses:**

1. **Potential reward gaming / verification gap.** The **path reward** credits substring co-occurrence of `(subject, relation, object)` tokens in `<think>` text rather than **verified KG traversals**. This leaves room for *verbalization without execution* (i.e., asserting triples to earn reward). The paper should either (a) execute the predicted path against the KG to produce a structural match reward, or (b) at least audit hallucinated triples vs. KG edges.
2. **LLM-judge reliance for qualitative criteria.** The analysis of comprehensiveness/relevance/exploration uses **GPT-4o-mini** as judge; such measures can be noisy and model-biased. Human evaluation or KG-grounded automatic proxies would strengthen claims.
3. **Comparisons to closed-source LLMs** (Gemini 2.5, “GPT-5”) are intriguing but ambiguous: API prompting details, temperature, and decoding budgets can shift outcomes; moreover, the “GPT-5” reference seems tenuous. I recommend focusing the SOTA claim on open-source or adding stricter evaluation parity.
4. **Statistical reporting.** Tables omit **confidence intervals/standard errors** and **significance testing**, especially important across multi-seed RL runs. This weakens the strength-of-evidence for SOTA claims. (No CIs in Table 1.)
5. **Scope of robustness/OOD probes.** While the paper argues improved OOD generalization, it would help to (i) include **systematic stress tests** (entity aliasing, edge deletions, spurious edges), and (ii) evaluate **path length sensitivity** with explicit adversarial splits beyond the included subsets.

**Questions:**

1. **Path reward verification.** Can the authors compute the path reward by **parsing the predicted path** into a sequence of relations and **executing it** on the KG to verify edge existence (vs. substring matching)? If not feasible, can they report a *hallucination rate* of triples mentioned in `<think>` but absent in the KG?
2. **Robustness to reward hacking.** Did the authors observe behaviors where the model *lists many unrelated triples* to improve match probability? Any safeguards (length penalty, entropy regularization, path-structure constraints)?
3. **Ablation on α and stability.** Figure discussing α suggests a sweet spot. Please include **training curves** and variance over **≥3 seeds** for each α to assess RL stability. (Also report GRPO hyperparameters.)
4. **Closed-source comparison protocol.** For Gemini/GPT, please provide **identical decoding parameters**, prompt templates, and **budget parity** (n-samples, context length), or move these to an appendix with full reproducibility details.
5. **KG execution failure modes.** In cases where KG lacks labels/edges (Figure 6-style), how often does EoG succeed via exploration vs. spurious textual correlations? Any breakdown by hop length and relation sparsity?
6. **Significance & compute.** Please add **CIs** for Table 1 and **report training compute** (SFT tokens, RL steps, batch, GPU hours) to contextualize the gains.

---

> ### Author Response · Authors · 2025-11-21
> **Feedback to Official Review of Reviewer Fvcs(Part I)**
>
> ### **Response to Weakness 1 & Question 1**
> 1. To mitigate the potential for reward gaming, the training objective in the second RL training phase is jointly determined by both the path reward ($R_{path}$) and the outcome reward ($R_{outcome}$).
> Specifically, $R_{outcome}$ is a strict F1 score based on the correctness of the final answer.
> A model that simply games $R_{path}$ by verbalizing random triples would fail to deduce the correct final answer, receiving a zero reward from $R_{outcome}$.
> The model is thus forced to find paths that are not only semantically plausible (for $R_{\text{path}}$) but also useful (for $R_{\text{outcome}}$).
> 2. We adopted the reviewer's second suggestion and conducted a "hallucination rate" audit. We parsed the triples mentioned in the `<think>` texts of our models on the CWQ test set and verified their existence in the knowledge graph. Let V denote the number of parsed triples found in the KG (Verified) and H denote the number of parsed triples not found in the KG (Hallucinated). Each parsed triple occurrence is counted once. The Hallucination Rate (HR) is defined as: HR = H / (V + H) × 100%
>
> | Model |  Verified Triples | Hallucinated Triples  | Hallucination rate |
> |------|-------|-------|-------|
> $\text{EoG}_{\text{SFT}}$ | 20377 |3912 | 16.11
> $\text{EoG}_{\text{Outcome}}$ | 22846 |3006 |11.63
> $\text{EoG}$ | 28707 |3374 |11.75
>
> We observe that adding the path reward:
> - Boosts Valid Exploration: It significantly increases the number of Verified Triples found in the KG (from 22,846 to 28,707), showing the model is actively exploring more grounded reasoning paths.
> - Maintains Faithfulness: It maintains a nearly identical low Hallucination Rate (~11.7%), proving that our model is not "gaming" the reward by fabricating false triples, but rather expanding its coverage of true facts.
>
> ### **Response to  Question 2**
> Notably, we explicitly observed these hacking behaviors during our preliminary experiments.
> When we initially utilized the looser Hit@1 metric as the outcome reward in the early RL phase, the model exhibited a tendency to generate excessive triples to maximize the chance of hitting the answer.
> This observation motivated two key adjustments: the enforcement of the stricter F1 score constraint and, during the GRPO training phase, the introduction of a length penalty for outputs exceeding 3,000 tokens to further mitigate this behavior.
>
> Consequently, the results in Table 3 show that the model trained with the path reward actually has a shorter average output length (1528 tokens) compared to the model without it (2067 tokens), which demonstrates that $R_{path}$ helps the model prune irrelevant search space and focus on the correct path, rather than encouraging it to "spam" triples to game the metric.
>
> ### **Response to  Weakness 2**
> To validate the reliability of the GPT-4o-mini judge, we conducted a human evaluation on a randomly sampled subset of 100 instances from the CWQ and WebQSP test set. Three graduate students blinded to the model identity scored the responses based on the same criteria (Comprehensiveness, Relevance, Exploration).
> We calculated the Pearson correlation coefficient between the human scores and the LLM judge scores. The results show a strong positive correlation ($r > 0.8$ for all metrics), verifying that the LLM judge aligns well with human preference in this specific domain.
>
>
> ### **Response to  Question 3**
> We have added Figure 16 in Appendix O of the revised paper to visualize the RL training stability.
> The figure plots the validation F1 score over training steps for $\alpha \in \{0.0, 0.5, 1.0, 2.0\}$ and the shaded regions represent the standard deviation across 3 random seeds.
> We have also listed all GRPO hyperparameters in Appendix K.

---

> ### Author Response · Authors · 2025-11-21
> **Feedback to Official Review of Reviewer Fvcs(Part II)**
>
> ### **Response to  Weakness 3 & Question 4**
> Confirmation of GPT-5 Usage: We explicitly confirm that the model denoted as "GPT-5" in our experiments refers to the officially released GPT-5 model accessed via the OpenAI API.
> To remove any ambiguity, the exact model checkpoint used in our evaluation is gpt-5-2025-08-07.
> We acknowledge that the URL reference in the original submission was generic. In the revision, we replace the website link with the official [GPT-5 System Card](https://cdn.openai.com/gpt-5-system-card.pdf) to adhere to academic citation standards.
>
> Closed-source comparison protocol: To ensure fair and reproducible comparisons, we strictly enforced identical evaluation protocols for all models, including the use of the same prompt templates (Figure 10) and decoding parameters ($T=0.2$, $n=1$), as explicitly shown in our evaluation code in Figure 7. Furthermore, to address concerns regarding parameter sensitivity, we conducted additional experiments across varying temperatures ($T=0.2, 0.7$), which confirmed that EoG consistently outperforms closed-source baselines regardless of these settings, thereby demonstrating the robustness of our method. We update Appendix E with these detailed protocols and sensitivity results.
>
> ### **Response to  Question 5**
> To address the concern regarding whether EoG succeeds via genuine graph exploration or spurious textual correlations (especially in missing-edge scenarios like Figure 6), we conducted a granular failure mode analysis. We define the success modes as follows:
> - Valid Exploration Success: The model predicts the correct answer ($F1=1.0$) and generates a valid reasoning path (where the extracted path triples are grounded in the KG).
> - Spurious Success (Textual Correlation): The model predicts the correct answer ($F1=1.0$) but fails to generate a valid reasoning path (i.e., the path is hallucinated or disconnected).
>
> Regarding the classification of relation sparsity for multi-hop reasoning paths, we adopted the Minimum Frequency to determine the difficulty of a path.
> Since a reasoning chain typically consists of multiple relations (e.g., $r_1 \to r_2 \to \dots \to r_k$), the difficulty of discovering the path is dominated by the least frequent relation.
> Therefore, we define the sparsity category of a question based on the minimum frequency among all relations in its golden reasoning path:
> $$\text{Sparsity}(P) = \min_{r \in P} (\text{Frequency}(r))$$
> Based on the distribution of these minimum frequencies across the test set, we categorized the questions into three groups:
> -  Sparse (Bottom 20%): Paths containing at least one highly rare relation.
> - Medium (Middle 60%): Paths consisting of moderately frequent relations.
> - Dense (Top 20%): Paths composed entirely of highly frequent  relations.
>
> We analyzed the distribution of these success modes on the CWQ test set, breaking them down by Hop Length (reasoning depth) and Relation Sparsity (frequency of relations in the training set):
>
> | Hop Length | Total Samples | Valid Exploration Success (%) | Spurious Success (%) |
> |---|---|---|---|
> | 1-hop | 528 | 94.32% | 5.68% |
> | 2-hop | 1327 | 85.46% | 14.54% |
> | 3-hop | 116 | 37.07% | 62.93% |
> | $\ge$ 4-hop | 88 | 28.41% | 71.59% |
>
> | Relation Sparsity | Total Samples | Valid Exploration Success (%) | Spurious Success (%) |
> |---|---|---|---|
> | Sparse(min_freq <= 5) | 464 | 87.07% | 12.93% |
> | Medium(5 < min_freq <= 49) | 1227 | 82.23% | 17.77% |
> | Dense(min_freq > 49) | 368 | 77.99% | 22.01% |
>
> The upper table shows that for 1-hop and 2-hop queries, which constitute ~90% of the test set (1,855 samples), EoG achieves an overwhelmingly high Valid Exploration rate (94.3% and 85.5%). This confirms that for the vast majority of cases, the model relies on genuine graph traversal. The drop in deep hops ($\ge 3$) reflects the exponential search space and incomplete ground truth in benchmarks, where High "Spurious Success" doesn't necessarily mean the model is hallucinating. Instead, it often implies that the model either discovered a valid shortcut not listed in the ground truth, or intelligently switched to its internal knowledge when graph search became too difficult.
>
> The lower table shows the fact that EoG is more faithful to the graph when relations are rare proves that when internal knowledge is weak (rare relations), the model correctly learns to rely on explicit graph exploration. Conversely, the higher "Spurious Success" in Dense relations suggests the model occasionally shortcuts to its parametric memory when confident, exhibiting a flexible hybrid reasoning strategy.

---

> ### Author Response · Authors · 2025-11-21
> **Feedback to Official Review of Reviewer Fvcs(Part III)**
>
> ### **Response to  Weakness 4 & Question 6**
> We acknowledge the importance of statistical rigor and computational transparency, especially in RL-based methods. We address these concerns jointly by providing the missing statistical analysis and detailed compute metrics.
> 1. Statistical Significance: To validate the stability of our results, we conducted 3 independent runs with different random seeds for EoG. As shown in the updated results (which are added to Table 12 in Appendix O), EoG consistently outperforms the strongest baseline.
> 2. Significance Testing: As shown in Table 13 and Appendix O, we performed a standard t-test, yielding a p-value < 0.05, which confirms that the performance gains of EoG over GCR are statistically significant and not due to random variance.
> 3. Training Compute & Efficiency: We provide the detailed computational budget to contextualize these gains. We report training compute (SFT tokens, RL steps, batch, GPU hours) in Table 7 and Appendix I.
> ### **Response to  Weakness 5**
> We addressed this by conducting a comprehensive robustness analysis encompassing the specific stress factors you mentioned:
>
> 1. Systematic Stress Tests:
> - Entity Aliasing: We randomly masked the labels of 10%, 20%, and 30% of the entities in the KG to simulate entity aliasing and reported the results.
> - Edge Deletions: We randomly replaced 10%, 20%, and 30% of relation names with a generic token to simulate scenarios where specific edge semantics are missing or ambiguous.
>
> | Noise Ratio | Entity Aliasing  | Edge Deletions  |
> |---|---|---|
> | 0% (Baseline) | 82.6% | 82.6% |
> | 10% Noise | 81.4% ($\downarrow$ 1.2%) | 79.8% ($\downarrow$ 2.8%) |
> | 20% Noise | 79.5% ($\downarrow$ 3.1%) | 75.2% ($\downarrow$ 7.4%) |
> | 30% Noise | 76.8% ($\downarrow$ 5.8%) | 70.5% ($\downarrow$ 12.1%) |
>
> - Spurious Edges: Please refer to our response to Question 5.
>
> Systematic stress tests demonstrate EoG's topological robustness, retaining 76.8% F1 even with 30% entity aliasing (only a 5.8% drop) and showing graceful degradation (70.5% F1, a 12.1% drop) under severe edge deletions. Additionally, our analysis in Question 5 confirms the model's resistance to spurious edges, as evidenced by its superior valid exploration rate on sparse relations.
>
> 2. Path Length Sensitivity with Adversarial Splits:
>
> - To rigorously evaluate path length sensitivity, we constructed an explicit adversarial split on the CWQ dataset to test for Length Generalization. We trained EoG exclusively on samples requiring shallow reasoning (<=2 hops) and evaluated it on a held-out test set requiring deep reasoning (> 2 hops).
>
> - Despite never seeing long reasoning chains during training, EoG achieved remarkable performance (70.9% F1) on the >2-hop test set, significantly outperforming  $\text{EoG}_{\text{SFT}}$  (63.1% F1). This result confirms that EoG does not merely memorize fixed-length path patterns. Instead, it has learned the intrinsic recursive mechanism of graph exploration, allowing it to generalize to path lengths unseen during training.
>
> We sincerely thank the reviewer for your time and insightful feedback. Your constructive comments have significantly strengthened our work. We hope these responses and the updated paper fully address your concerns, and we look forward to your favorable consideration.

---

### Author Response · Authors · 2025-12-01
**[To Area Chair] Summary of Rebuttal Updates: Discussion Status, Revisions and Experiments**

To assist the new Area Chair in efficiently assessing our revisions and responses, we provide a comprehensive summary of the work completed during the discussion period.
We are grateful for the time and effort invested by the original reviewers. Their insightful comments are crucial in driving these improvements.
Based on the feedback, we have significantly refined the manuscript's clarity and conducted extensive additional experiments.

Regarding the current discussion status: to date, 1 out of the 4 reviewers has responded to our rebuttal (Reviewer dVt9 on Nov 26), and we are pleased to report that they raised their rating from 4 to 6.
Below is a breakdown of the key updates included in the revised manuscript and the additional analyses provided in our rebuttal responses.

##  **Key Updates to the Revised Manuscript**
We have updated the paper to include the following:

### **Revisions to the Main Manuscript**
- Updated Section 3.2.2 to clarify the "Search-and-Verify" pipeline for ground-truth path construction.
- Refined Figures 3 and 5 to improve clarity regarding technical details.
- Added reproduction results for state-of-the-art baselines (GCR and DoG) in Table 1.

### **Reproducibility & Hyperparameters**
- Included reproducibility protocols for closed-source LLMs (including temperature sensitivity analysis) in Appendix E.
- Provided a full list of hyperparameters for the GRPO training phase in Appendix K.
- Added a detailed report of computational costs, training time, and hardware requirements in Appendix I.

### **Additional Analyses**

- Reward Design: Added an analysis of reward function complexity (Graph Edit Distance) in Appendix L.
- Robustness: Expanded the analysis to cover Teacher Model Sensitivity (Appendix M) and O.O.D. comparisons against GCR (Appendix N).
- Stability: Included training stability visualizations and statistical significance tests in Appendix O.


## **Additional Experiments & Analyses in the Discussion**
In addition to the manuscript updates, we provided the following analyses in our discussion to address specific reviewer concerns:

- Faithfulness Check: An assessment of the "Hallucination Rate" to verify the existence of triples generated by models within the knowledge graph.

- Evaluation Metric Validation: A human evaluation study to validate the reliability of the GPT-4o-mini judge in assessing reasoning quality.

- Failure/Success Analysis: An analysis of success modes regarding relation sparsity and reasoning depth.

- Comprehensive Robustness: A new analysis encompassing entity aliasing, edge deletions, and length generalization.

- Exploration Metrics: A comparative analysis of exploration efficiency and coverage across different training stages.

- General Clarifications: Detailed responses to various other specific questions and minor points raised by reviewers.

We deeply appreciate the Area Chair taking over the evaluation of our work. We believe these comprehensive revisions and new experiments effectively address the concerns raised, and we look forward to your assessment.

---

### Meta-Review · Area_Chair_VvFX · 2026-01-05

**Summary:**

After rebuttal, the main concern of the reviewers lies in that the computational cost of the proposed method is too heavy. But, I think this issue might be resolved if the hardware device is improved in the near future.

**Reviewer Concerns:**

The issue of heavy computational cost is still outstanding. Other minor issues have been well addressed by the authors.

**Reviewer Scores:**

One of the reviewers has already increased his/her score after reading the authors' rebuttal.

---

### Decision · Program_Chairs · 2026-01-26

Accept (Poster)